# A unified framework to model synaptic dynamics during the sleep–wake cycle

Fukuaki L. Kinoshita[1,2], Rikuhiro G. Yamada[2,3], Koji L. Ode[4], Hiroki R. Ueda[2,3,4]*

1 Department of Neurology, Graduate School of Medicine, Osaka University, Osaka, Japan, 2 Laboratory for Synthetic Biology, RIKEN Center for Biosystems Dynamics Research, Osaka, Japan, 3 Department of Systems Biology, Institute of Life Science, Kurume University, Fukuoka, Japan, 4 Department of Systems Pharmacology, Graduate School of Medicine, The University of Tokyo, Tokyo, Japan

* uedah-tky@umin.ac.jp

**Academic editor:** Jozsef Csicsvari, Institute of Science and Technology Austria, AUSTRIA

## Abstract

Understanding synaptic dynamics during the sleep–wake cycle in the cortex is crucial yet remains controversial. The synaptic homeostasis hypothesis (SHY) suggests synaptic depression during non-rapid eye movement (NREM) sleep, while other studies report synaptic potentiation or synaptic changes during NREM sleep depending on activities in wakefulness. To find boundary conditions between these contradictory observations, we focused on learning rules and firing patterns that contribute to the synaptic dynamics. Using computational models considering mammalian cortical neurons, we found that under Hebbian and spike-timing dependent plasticity (STDP), wake-like firing patterns decrease synaptic weights, while sleep-like patterns strengthen synaptic weights. We refer to this tendency as Wake Inhibition and Sleep Excitation (WISE). Conversely, under Anti-Hebbian and Anti-STDP, synaptic depression during NREM sleep was observed, aligning with the conventional synaptic homeostasis hypothesis. Moreover, synaptic changes depended on firing rate differences between NREM sleep and wakefulness. We provide a unified framework that could explain synaptic homeodynamics under the sleep–wake cycle.

## Introduction

During wakefulness, organisms perceive external worlds through the five senses to learn and take appropriate actions. During sleep, organisms disconnect from the environment to reorganize memory and recover from fatigue. Recent studies have revealed that cortical neurons are responsible for these brain functions underlying learning and memory formation [1] and that the dynamics of the cortical synaptic weights are associated with the sleep–wake cycle [2,3]. A hypothesis known as the synaptic homeostasis hypothesis (SHY) proposed that wakefulness potentiates synapses through learning with the costs of higher energy demand while the sleep state depresses less important synapses to restore synaptic homeostasis [2]. However,

**Data availability statement:** Records for spike patterns of rats during the sleep–wake cycle are publicly available through CRCNS (Collaborative Research in Computational Neuroscience, http://dx.doi.org/10.6080/k02n506q). The code used in the article has been deposited and made publicly available on https://doi.org/10.5281/zenodo.15210933. All other relevant data are within the paper and its Supporting information files.

**Funding:** This study was supported by JST ERATO (grant no. JPMJER2001, https://www.jst.go.jp/erato/en/about/index.html) to H.R.U., Brain/MINDS (grant no. JP21dm0207049, https://brainminds.jp/en/) to H.R.U., the Science and Technology Platform Program for Advanced Biological Medicine (grant no. JP21am0401011, https://www.amed.go.jp/en/program/list/11/01/001.html) to H.R.U., Moonshot R&D (grant no. JPMJMS2023, https://www.jst.go.jp/moonshot/en/index.html) to R.G.Y, a Grant-in-aid for scientific research (S) (grant no. JP18H05270, https://www.jsps.go.jp/english/e-grants/) to H.R.U., a Grant-in-Aid from the Human Frontier Science Program (grant no. RGP0019/2018, https://www.hfsp.org/funding/hfsp-funding/research-grants) to H.R.U., MEXT QLEAP (grant no. JPMXS0120330644, https://www.jst.go.jp/stpp/q-leap/en/index.html) to H.R.U., intramural Grant-in-Aid from the RIKEN BDR to H.R.U., RIKEN Junior Research Associate Program (https://www.riken.jp/en/careers/programs/jra/) to F.L.K. The funders had no role in study design, data collection and analysis, decision to publish, or preparation of the manuscript.

**Competing interests:** The authors have declared that no competing interests exist.

**Abbreviations:** CaMKII, calcium/calmodulin-dependent protein kinase II; CV, coefficient of variance; DOWNM, mean down-state duration; ERK, extracellular signal-regulated kinase; NREM, non-rapid eye movement; ISI, inter-spike interval; ISIM, mean inter-spike interval; PCA, principal component analysis; SHY, synaptic homeostasis hypothesis; SIK3, salt-inducible kinase 3; SNR, signal to noise ratio; SSE, sum of squared errors; STDP, spike-timing dependent plasticity; SWO, slow-wave oscillation; UPM, mean Up-state duration; WISE, wake inhibition and sleep excitation.95% CI95% credible interval

the dynamics of synaptic weights in the sleep–wake cycle, especially during sleep, remain controversial. Several studies have demonstrated that NREM sleep potentiates synapses, contributing to memory consolidation. Others have reported extended wakefulness or sleep deprivation results in a loss of spines or reduced excitability in some brain regions [4–7]. Furthermore, SHY suggests that synapses strengthened during wakefulness are less susceptible to synaptic depression during NREM sleep [2]. In contrast, other studies propose the normalization of neuronal activities, where fast-firing neurons and slow-firing neurons during wakefulness are weakened and strengthened, respectively, during NREM sleep [8]. We sought to find the boundary conditions that reconcile these discrepancies and to comprehensively understand the sleep–wake synaptic dynamics.

The heterogeneity of brain states can confound in vivo studies because the slow-wave oscillation (SWO), that is a characteristic firing pattern of NREM sleep, also occurs in wakefulness and sleep states include REM sleep, which has wake-like firing patterns [9,10]. To address this issue, we developed computational models to investigate the direct relationships between synaptic weights and neuronal firing patterns characteristic of each brain state. To account for the diversity of neurons and obtain more robust conclusions that generalize across brain regions and even different species, computational simulations were conducted with different types of models and numerous randomly generated parameters. We prepared sleep- and wake-like firing patterns based on in vivo experiments, and devised a unified function that recapitulates typical synaptic learning rules in the cortex for updating the synaptic weights [11]. These settings allowed us to simulate the dynamics of synaptic weights under specific types of spike trains, such as burst firing, based on synaptic learning rules [12]. The synaptic learning rules we studied include the Hebbian rule, STDP, and their reverse types (Anti-Hebbian and Anti-STDP). According to the Hebbian rule, a synaptic connection between two neurons strengthens when pre- and post-synaptic neurons fire simultaneously [13]. A temporally asymmetric form of Hebbian rule is STDP. The classical STDP describes that synaptic potentiation occurs when pre-synaptic spikes precede post-synaptic spikes within a certain temporal window, while synaptic depression occurs in post-synaptic spikes precede pre-synaptic spikes [14,15]. Reverse types of these (Anti-Hebbian and Anti-STDP) are also observed in the mammalian cortex [16]. Such learning rules lead to synaptic depression when synapses are presented with correlated activity, serving critical functions in the discrimination of specific spike sequences and the detection of novel stimuli [17–20].

Our simulations revealed that synaptic weights become higher in sleep-like synchronized states than in wake-like desynchronized states under Hebbian and classical STDP, assuming the same mean firing rates for both sleep- and wake-like firing patterns. We refer to these dynamics as Wake Inhibition and Sleep Excitation (WISE). In contrast, synaptic depressions during sleep-like firing patterns, which represents SHY, were observed under Anti-Hebbian and Anti-STDP. Moreover, our results suggested that the synaptic dynamics also depend on mean firing rates, providing a unified framework for the synaptic homeodynamics of neural networks during the sleep–wake cycle.

## Results

### WISE under Hebbian and STDP, SHY under anti-Hebbian and anti-STDP

To investigate synaptic dynamics in NREM sleep and wakefulness with synaptic learning rules, we used a $Ca^{2+}$-based plasticity model. Graupner and colleagues proposed that the $Ca^{2+}$-based plasticity model with two thresholds for post-synaptic $Ca^{2+}$ can describe the various types of synaptic learning rules [21]. Based on Graupner's model, we developed a modified $Ca^{2+}$-based plasticity model to represent four different types of learning rules (Hebbian, STDP, Anti-Hebbian, and Anti-STDP) by setting eight parameters ($\theta_p$: potentiation threshold, $\theta_d$: depression threshold, $\gamma_p$: potentiation amplitude, $\gamma_d$: depression amplitude, $\tau_{pre}$: time constant for $Ca^{2+}$ from N-methyl-D-aspartate receptor (NMDAR), $\tau_{post}$: time constant for $Ca^{2+}$ from Voltage-gated $Ca^{2+}$ channel (VGCC), $\sigma$: amplitude for noise, and $\tau_s$: time constant for synaptic change) (Fig 1A). Synaptic weights were defined as being linearly related to synaptic efficacy ($\rho$) as $w = w_0 + \rho(w_1 - w_0)$, where $w_0$ and $w_1$ are minimum and maximum synaptic weights, respectively. $\rho$ is described by a first order differential equation according to the previous article ("Materials and methods, Modeling synaptic learning rules") [21]. We confirmed that our model could predict the experimentally observed changes of post-synaptic $Ca^{2+}$ and synaptic strength under stimulations with different time lags (Fig 1B and 1C). We randomly generated more than 1 million parameter sets and selected 1,000 parameter sets that well represent either one of four learning rules (Fig 1D and 1E, "Materials and methods, Parameter search for synaptic learning rules"). Each learning rule has a clear cluster in the distribution of thresholds and amplitudes (Fig 1F). The distributions of other parameters and those in other fitting conditions are shown in S1 Fig. To evaluate the change of synaptic weights during sleep-like and wake-like firing patterns, we assumed one post-synaptic neuron connected with 10 pre-synaptic neurons and the same mean firing rates both in sleep-like and wake-like patterns (Fig 1G). Sleep-like and wake-like firing patterns were derived from previous in vivo recordings [8,22] ("Materials and methods, Generation of sleep and wake-like spike patterns" and S18 Fig). Then, time changes in synaptic efficacy during the sleep-like and wake-like firing patterns were calculated and compared under synaptic learning rules. The mean synaptic efficacy became higher in sleep-like states than in wake-like states under Hebbian and STDP, representing WISE (Fig 1H). The opposite results were observed in Anti-Hebbian and Anti-STDP, representing SHY. Thus, WISE and SHY are observed under the specific types of learning rules, with both states exhibiting equal mean firing rates. Hereafter, we focus on relative net changes of synaptic efficacy in a local network, and we define the decrease of the efficacy as SHY and the increase as WISE during the sleep-like state.

### Robustness of WISE and SHY

Next, we investigated the robustness of the WISE and SHY in different settings. We first modified the previous model with one post-synaptic neuron connected by 10 pre-synaptic neurons to have 96 pre-synaptic neurons or random connections. This modified model still exhibited WISE under Hebbian and STDP, while it showed SHY under Anti-Hebbian and Anti-STDP (Fig 2A and 2B). Next, we tested modified parameters of time constants and amplitudes of learning rules to see if WISE and SHY depend on the properties of learning rules. Even in those settings, we found that these trends still held (Figs 2C and S3). In Fig 2D and 2E, we generated various sleep-like firing patterns by changing parameters such as the $\log_{10}(mean\ inter\text{-}spike\ interval)$ (ISIM), $\log_{10}(mean\ Up\text{-}state\ duration)$ (UPM), and $\log_{10}(mean\ Down\text{-}state\ duration)$ (DOWNM). We tested ranges of parameters for each targeted mean firing rate by changing either DOWNM or ISIM (Fig 2D and 2E). We found that the mean synaptic efficacy was higher in sleep-like states with most of the parameters in the tested range. Notably, this trend was more apparent at lower firing rates. These results validated that WISE and SHY were robust under various biologically feasible conditions.

### WISE and SHY in Hodgkin–Huxley-based network models

We then tested whether WISE and SHY hold in a more realistic setting where the synaptic efficacy can change the firing pattern. To recapitulate the variable firing pattern, we introduced the Hodgkin–Huxley model to network models based on

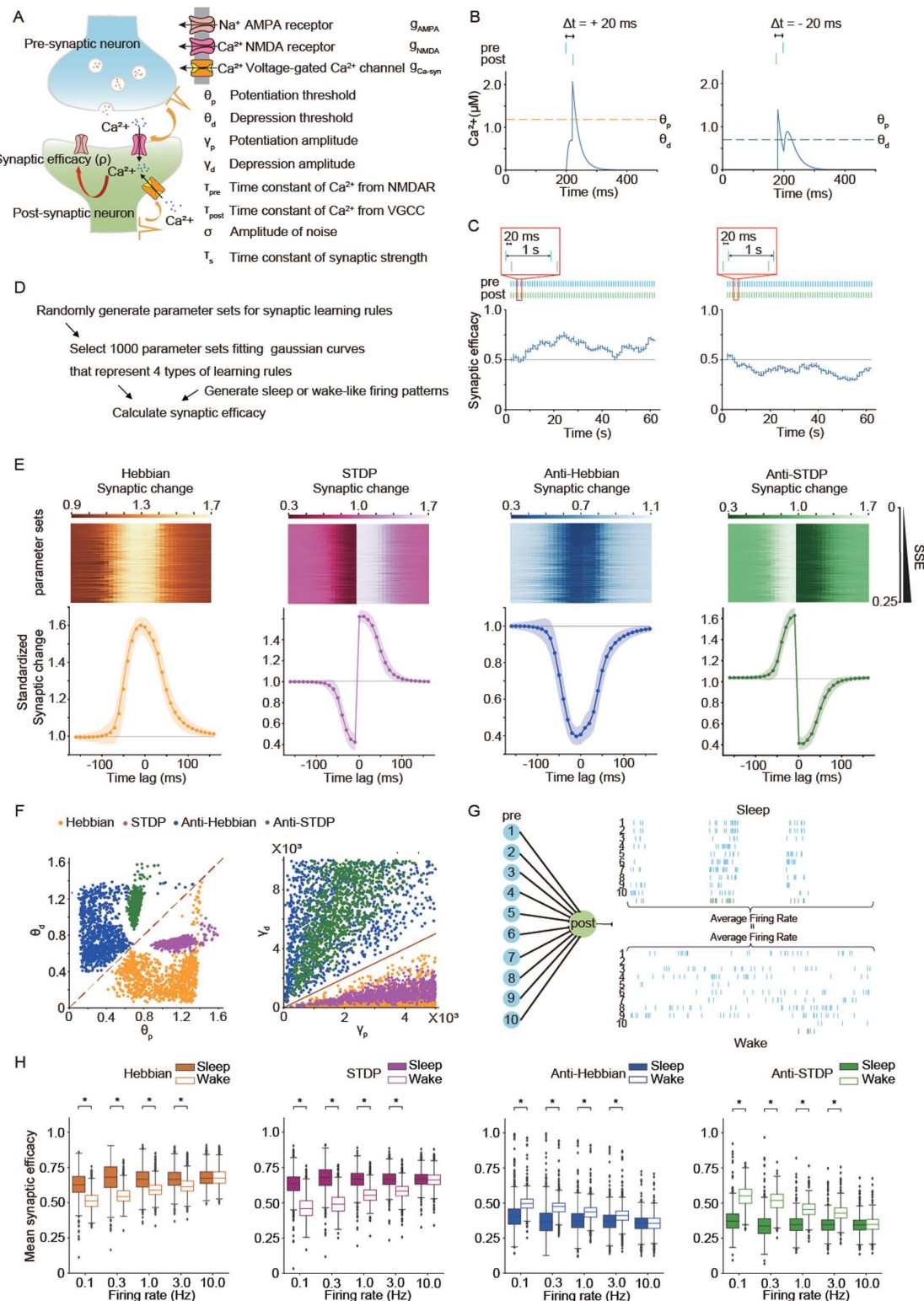

**Fig 1. WISE under Hebbian and STDP, SHY under Anti-Hebbian and Anti-STDP assuming the same firing rates during sleep-like and wake-like firing patterns. (A)** Schematic illustration of the Ca²⁺-based plasticity model for synaptic learning rules. Ca²⁺ in a post-synaptic neuron was calculated by summing Ca²⁺ from NMDAR and VGCC. Synaptic efficacy ($\rho$) between neurons was updated by the concentration of Ca²⁺ in a post-synaptic neuron. **(B)**

The simulated concentration of $Ca^{2+}$ in a post-synaptic neuron where a pair of stimulations with a short delay (20 ms) were given to two neurons under STDP. Two neurons represented by the equation (Eq. 1–10) were connected by a single unidirectional synapse. The parameters for STDP are shown in S8 Table. **(C)** The simulated dynamics of synaptic efficacy where two neurons were repeatedly stimulated at 60 Hz with a short delay (20 ms) under STDP. Two neurons represented by the equation (Eq. 1–10) were connected by a single unidirectional synapse. Synaptic weights were defined as being linearly related to synaptic efficacy ($\rho$) as $w = w_0 + \rho(w_1 - w_0)$, where $w_0$ and $w_1$ are minimum and maximum synaptic weights, respectively. The parameters for STDP are shown in S8 Table. **(D)** The procedure of parameter search for synaptic learning rules and calculation of synaptic efficacy during sleep-like and wake-like firing patterns. **(E)** The standardized synaptic changes of 1,000 parameter sets collected by parameter search for four types of learning rules. Two neurons represented by the equation (Eq. 1–10) were connected by a single unidirectional synapse and stimulated at 60 Hz for 1 min with different time lags under a parameter set for a synaptic learning rule. Initial synaptic efficacy was 0.5, and the synaptic change in each time lag was calculated as mean synaptic efficacy after the stimulation protocol divided by the initial synaptic efficacy (synaptic efficacy before stimulation). A total of 1,000 parameter sets whose sum of squared errors (SSE) between the analytical result and fitted Gaussian curves was less than 0.25 were selected. Each row of the upper panel represents the numerical values of synaptic changes (after/before) in different lag times, computed using parameters that were arranged in ascending order based on their SSE. The line and shadow in the lower panel indicate the mean and standard deviation, respectively. **(F)** Distributions of 1,000 parameter sets collected by parameter search for each learning rule in the axes of thresholds ($\theta_p$ and $\theta_d$) and amplitudes ($\gamma_p$ and $\gamma_d$). **(G)** Schematic illustration for connections and firing patterns of neurons used in the calculation of synaptic efficacy during sleep-like or wake-like firing patterns. Ten pre-synaptic neurons were connected to one post-synaptic neuron. All the neurons were represented by the equation (Eq. 1–10). The mean firing rates during sleep-like and wake-like firing patterns were adjusted to the same. **(H)** Box plots for mean synaptic efficacy in sleep-like and wake-like firing patterns by different synaptic learning rules and mean firing rates ($n = 1,000$ for each firing rate, $n$ represents the number of synaptic learning rules). A total of 1,000 parameter sets for the specific learning rules collected by parameter search in panel **(E)** and connection patterns in panel **(G)** were applied. Initial synaptic efficacies in all the synapses were 0.5 and synaptic efficacies were simulated for 6 min. Synaptic efficacies for the last 2 min were averaged and compared between sleep-like and wake-like firing patterns. The whiskers above and below show minimal to maximal values. The box extends from the 25th to the 75th percentile, and the middle line indicates the median. Bayesian statistical analysis was performed using Markov Chain Monte Carlo method to infer posterior distributions of average differences in mean synaptic efficacy between sleep-like firing patterns and wake-like firing patterns. Asterisks (*) indicate 95% credible intervals (CIs) do not include zero. The data underlying the graphs shown in the figure can be found in Table A in S1 Data. The 95% CIs for the distributions of average differences are shown in Table A in S3 Data.

our previous study [23,24], with a ratio of excitatory and inhibitory neurons of 4:1 (Fig 3A, see "Materials and methods, Hodgkin–Huxley-based network model"). In this study, we considered molecules responsible for generating SWO in three subcellular compartments (post-synaptic, intracellular (cell body), and pre-synaptic compartments) [23,25]. We assumed that NMDAR and VGCC function in post-synapses and cell bodies, respectively, for generating SWO [23]. In addition, because pre-synaptic transmission is crucial for generating SWO in the cortex [25], we modeled AMPAR, NMDAR, and GABAR to receive the neural transmission. Parameter searches for SWO and bifurcation analysis for a single neuron were based on previous articles [23]. First, we randomly generated parameter sets from a large parameter space within the biophysically feasible range and searched for parameter sets that yield firing patterns of SWO. Then, we searched for parameter sets that bifurcate from wake-like to sleep-like patterns as a network (Fig 3B, see "Materials and methods, Parameter search for SWO and bifurcation analysis in Hodgkin–Huxley-based network models"). Clear desynchronization and synchronization, evaluated by the coefficient of variance (CV) of spike counts per 50 ms (sleep score) (see "Materials and methods, Evaluation of synchronization and desynchronization in Hodgkin–Huxley-based network models"), were observed with parameter sets for each subcellular component (Figs 3C, 3D and S4). We then selected the parameter sets that showed almost the same mean firing rates in sleep-like and wake-like states and evaluated synaptic efficacy in the sleep-like and wake-like states (S6 and S7 Figs). WISE was observed under STDP (Fig 3E), while SHY was observed under Anti-STDP (Fig 3F). Similar trends were observed under Hebbian and Anti-Hebbian and in network models with different connections (S9 and S10 Figs). These results validated WISE and SHY in a realistic network model.

### WISE under Hebbian and STDP and SHY under Anti-Hebbian and Anti-STDP is compatible with models including sleep–wake dynamics

Next, we incorporated the spontaneous sleep–wake cycle into our network models. Previous phosphoproteomic studies suggested that phosphorylation of several synaptic proteins is associated with sleep needs [26–28]. The sleep needs increase during wakefulness and decreases with the onset of sleep. This homeostatic oscillation of sleep needs is referred to as

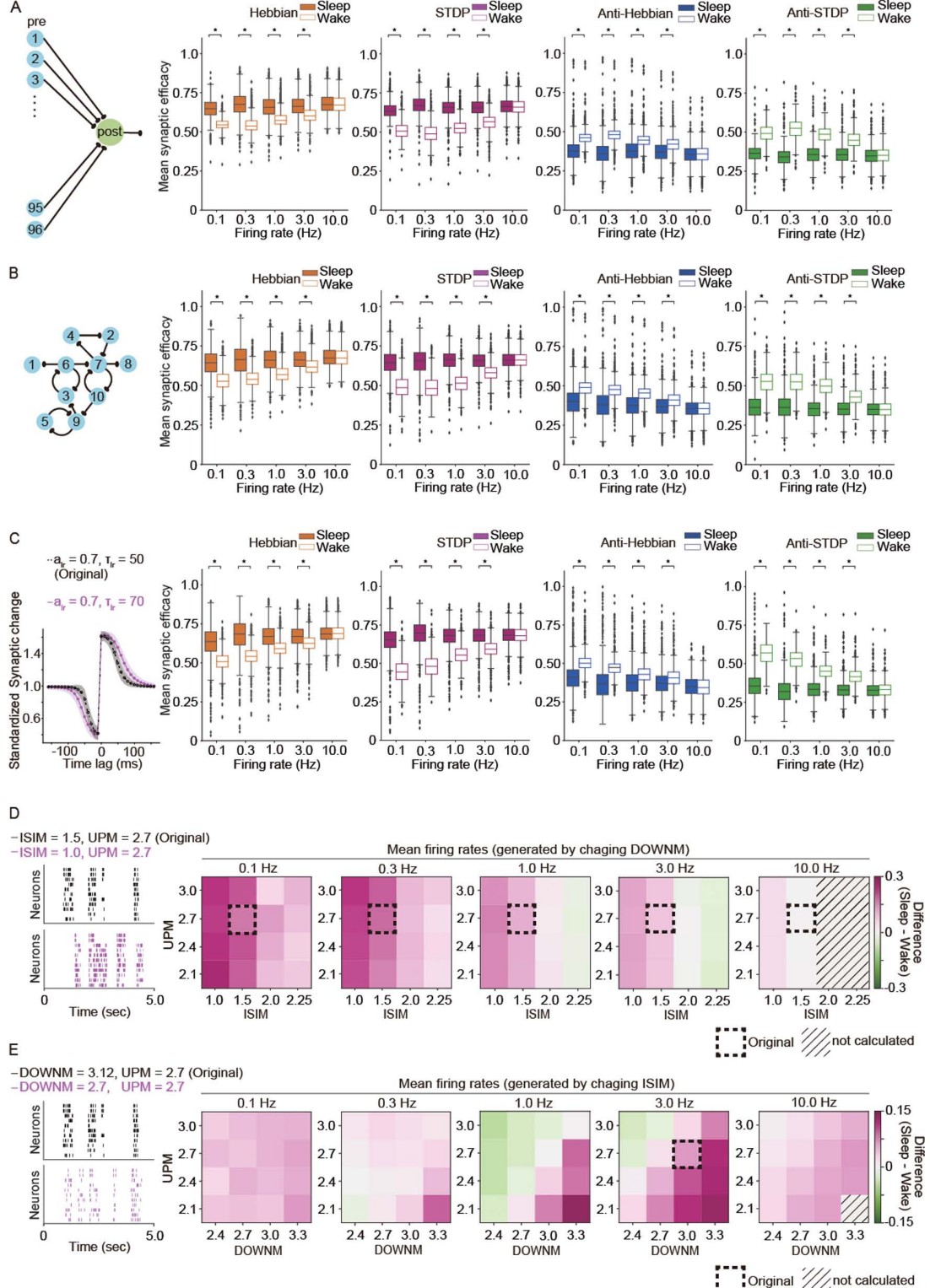

**Fig 2. Robustness of WISE and SHY.** Synaptic efficacies were calculated and compared between sleep-like and wake-like firing patterns in different conditions, assuming the same firing rates in both states. All the neurons were represented by the equation (Eq. 1–10). A total of 1,000 parameter sets for the specific learning rules collected by parameter search in Fig 1 were applied. Initial synaptic efficacies in all the synapses were 0.5, and synaptic

efficacies were simulated for 6 min. Synaptic efficacies for the last 2 min were averaged. **(A)** Schematic illustration of network model consisting of 96 pre-synaptic neurons connected to one post-synaptic neuron and box plots for mean synaptic efficacy during sleep-like and wake-like firing patterns. **(B)** Schematic illustration of network model consisting of 10 randomly connected neurons and box plots for mean synaptic efficacy during sleep-like and wake-like firing patterns. **(C)** The original (black dotted line) and modified (magenta solid line) curves for STDP rule and box plots for mean synaptic efficacy. $a_{lr}$ is the amplitude, and $\tau_{lr}$ is the time constant for Gaussian curves to be fitted by parameter search. **(D)** Representative raster plots of the original (black) and modified (magenta) sleep-like firing patterns and the differences in median of mean synaptic efficacies between sleep-like and wake-like firing patterns (*sleep – wake*). The adjusted mean firing rates of sleep-like firing patterns were generated by changing DOWNM under constant UPM and ISIM ($n$ = 1,000 for each firing rate). The dotted boxes highlighted the values of UPM = 1.5 and ISIM = 2.7, which were the original values used in Fig 1H. **(E)** Representative raster plots of the original (black) and modified (magenta) sleep-like firing patterns and the differences in median of mean synaptic efficacies between sleep-like and wake-like firing patterns (*sleep – wake*). The adjusted firing rates of sleep-like patterns were generated by changing ISIM under constant UPM and DOWNM ($n$ = 1,000 for each firing rate, $n$ represents the number of synaptic learning rules). The dotted boxes highlighted DOWNM = 3.0 and UPM = 2.7, which are the closest to the original values used in Fig 1H: DOWNM = 3.12 and UPM = 2.7. **(A–C)** The mean synaptic efficacy was evaluated in sleep-like and wake-like firing patterns with various mean firing rates assuming different synaptic learning rules ($n$ = 1,000 for each firing rate, $n$ represents the number of synaptic learning rules). The whiskers above and below of box plots show minimal and maximal values, respectively. The box extends from the 25th to the 75th percentile and the middle line indicates the median. Bayesian statistical analysis was performed using Markov Chain Monte Carlo method to infer posterior distributions of average differences in mean synaptic efficacy between sleep-like firing patterns and wake-like firing patterns. Asterisks (*) indicate 95% CIs do not include zero. The data underlying the graphs shown in the figure can be found in Tables B–D in S1 Data. The 95% CIs for the distributions of average differences are shown in Tables B–D in S3 Data. **(D, E)** ISIM: $\log_{10}$(*mean ISI*), UPM: $\log_{10}$(*mean Up-state duration*), DOWNM: $\log_{10}$(*mean Down-state duration*).

Process S [29]. In the present model, we assumed that calcium/calmodulin-dependent protein kinase II (CaMKII) is responsible for the homeostatic oscillation [30]. To evaluate the synaptic efficacy across a series of sleep–wake cycles, we assumed that CaMKII changes its states in a use-dependent manner during wakefulness and induces SWO by interacting with channels or receptors that regulate neuronal membrane potentials or enzymes that regulate neurotransmitters [10,30]. This assumption aligns with observations that CaMKII has multiple phosphorylation states and changes its function accordingly [31]. We integrated the use-dependent change of CaMKII's function for activating channels such as NMDAR into the sleep–wake dynamics model. The initial state of CaMKII, such as pT286/287 CaMKII, has self-activating ability and $Ca^{2+}$-dependent activation [32]. This initial state activates the second state of CaMKII, such as pT305/306 CaMKII, or phosphatases, such as calcineurin, which can be regulated during sleep [33]. We assumed that the second state directly interacts with molecules that induce SWO from the result of optimizing correlation to Process S for network models (S11A Fig).

In the simulation of a representative model, we observed that pT286/287 CaMKII, represented as *r* in Fig 4A, gradually increased due to $Ca^{2+}$ influx and autoactivation during wakefulness, which was followed by an increase in pT305/306 CaMKII, represented as *a* in Fig 4A. The increased *a* then activated NMDAR and induced SWO (Fig 4C and 4D). The mean synaptic efficacy was higher during sleep-like periods than during wake-like periods under STDP (Fig 4E and 4G). The opposite results were observed under Anti-STDP (Fig 4F and 4H). The differences of synaptic efficacy between sleep and waking states were observed in the analysis with multiple parameter sets (S11B and S11C Fig). The results of model with other bifurcation mechanisms or systems also showed the same trends (S12–S14 Figs). These results confirmed that WISE under STDP and SHY under Anti-STDP are compatible with network models that have sleep–wake dynamics.

### Synaptic changes depend on firing rates assuming higher firing rates during wake-like states

In the previous sections, we compared synaptic efficacies in sleep-like and wake-like states by assuming the same mean firing rates in both states. While this assumption is feasible in brain regions such as the visual cortex, where firing rates are almost constant between states [34], regions such as the somatosensory cortex showed higher firing rates during wakefulness [35]. To evaluate the synaptic dynamics in the higher firing rates during wake-like states, we assumed that mean firing rates in Up states of sleep-like patterns are equal to those in wake-like patterns and found that WISE was observed at lower mean firing rates while SHY was observed at higher mean firing rates under Hebbian and STDP (Fig 5). Thus, synaptic efficacies change to different directions depending on mean firing rates assuming higher firing rates during wake-like states.

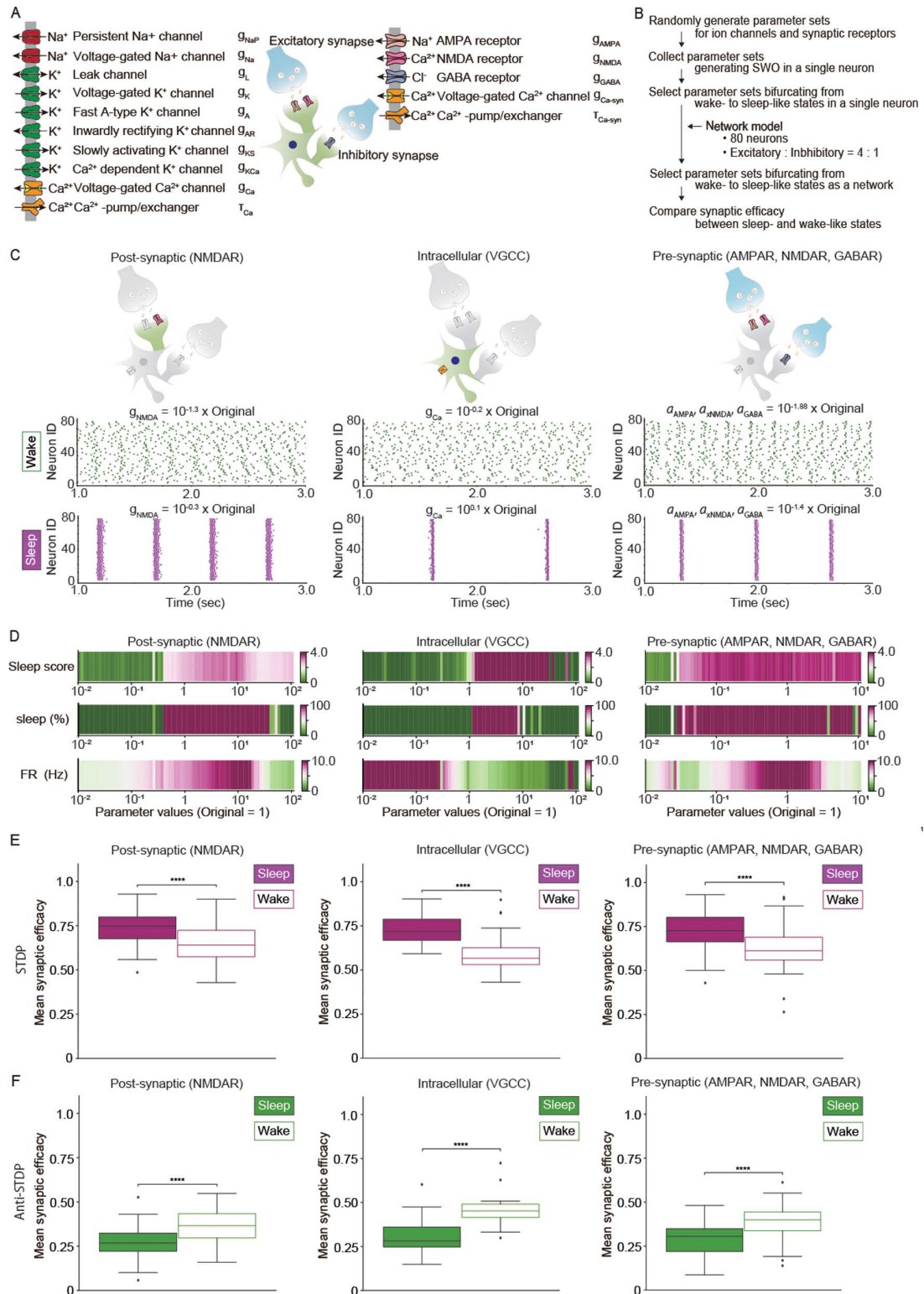

**Fig 3. WISE and SHY in Hodgkin–Huxley-based network models. (A)** Schematic illustration of the model for a neuron with an excitatory and inhibitory synapse. AMPAR, NMDAR, GABAR, and VGCC were defined as synaptic receptors or channels. $Ca^{2+}$-pump/exchangers were defined in cellular and synaptic compartments independently (represented by $\tau_{Ca}$ and $\tau_{Ca-syn}$, respectively). A Hodgkin–Huxley-based network model was constructed

based on the averaged neuron model in a previous study [23]. **(B)** Procedures for collecting parameter sets that bifurcate from wake-like to sleep-like firing patterns and comparing synaptic efficacy between the two states in Hodgkin–Huxley-based network models. Network models consisted of 80 neurons with E:I ratio of 4:1. Each neuron had two excitatory and two inhibitory synapses. Connections between inhibitory neurons were not considered. **(C)** Representative sleep-like and wake-like firing patterns for three types of bifurcation models. In the pre-synaptic models, firing patterns are bifurcated by the values representing pre-synaptic activation (see "Materials and methods, Parameter search for SWO and bifurcation analysis in Hodgkin–Huxley-based network models"). The parameters used in the simulation were obtained by multiplying the original values defined in S5 Table by the presented factors. Simulations were conducted for five seconds and firing patterns during three seconds are presented in each model. **(D)** Changes in the sleep score, percentage of sleep-like waveforms (sleep (%)), and mean firing rate (FR (Hz)) as the conductance of the channel or receptor, or the coefficients of pre-synaptic activations are gradually increased. The range of the conductance was divided into 80 steps for post-synaptic or intracellular bifurcation, while the range of the coefficient was divided into 75 steps for pre-synaptic bifurcation. Simulations were conducted for 10 s at each conductance or coefficient step. **(E, F)** Box plots for mean synaptic efficacy during sleep-like and wake-like firing patterns under STDP **(E)** and Anti-STDP **(F)** by three types of network models (n = 191, 52, and 150 for STDP and n = 121, 36, and 119 for Anti-STDP in post-synaptic, intracellular, and pre-synaptic bifurcation models, respectively. n represents the number of parameter sets for the network models). A parameter set for the synaptic learning rule was assigned to excitatory synapses in each network model. Synaptic efficacy was compared, assuming the firing rates between sleep-like and wake-like states were almost close. Initial synaptic efficacies of all synapses were 0.5. Simulations were conducted for 60 s, and synaptic efficacy and CV were averaged over a period of 10 to 60 s. The whiskers above and below of box plots show minimal and maximal values, respectively. The box extends from the 25th to the 75th percentile, and the middle line indicates the median. $*p < 0.05$, $**p < 0.01$, $***p < 0.001$, $****p < 0.0001$, Student $t$ test was applied. The data underlying the graphs shown in the figure can be found in Table E in S1 Data.

## Discussion

In this study, we investigated how synaptic dynamics interact with firing patterns and learning rules. Under Hebbian and STDP, wake-like firing patterns inhibits the synaptic connections, hence weakening the synaptic efficacy, while sleep-like firing patterns excite synaptic connections, hence strengthen the synaptic efficacy. We referred to this tendency as WISE. In contrast, under Anti-Hebbian and Anti-STDP, wake-like and sleep-like patterns tend to strengthen and weaken synaptic efficacies, respectively, which aligns with SHY. When we set the firing rate of the Up state of sleep-like phasic firing patterns equal to the firing rate of wake-like tonic firing patterns, the resulting higher firing rate of wake-like firing pattern tends to strengthen synapses. This indicates that firing rate is the dominant factor in determining the direction of synaptic changes. These findings delineate the boundary conditions of synaptic dynamics during the sleep–wake cycle. We also demonstrated that these boundary conditions are stable under various conditions by using two types of models with numerous parameters derived from biological knowledge.

### Boundary conditions in synaptic homeodynamics

From the perspective of homeostasis, SHY proposes that wakefulness strengthens synapses to learn about the environment, while NREM sleep weakens less important synapses to reduce energy consumption [2]. Although studies, including anatomical and electrophysiological research, support SHY [26,36,37], several studies have reported contradictory results [4,5,8,38].

Our study indicated that the direction of synaptic changes during sleep-like firing patterns depends on synaptic learning rules and firing rates of local networks. Assuming the same mean firing rates in sleep-like and wake-like patterns, SHY is observed under Anti-Hebbian and Anti-STDP (Figs 1H, 2A–2C, 3E, 3F, 4G and 4H). Additionally, we observed that higher mean firing rates results in smaller differences in synaptic changes between sleep-like and wake-like states (Figs 1H, 2A–2C, and S9). This finding suggests that neurons with higher firing rates during wakefulness are less susceptible to synaptic depression during sleep, consistent with SHY [2]. In contrast, we observed WISE under Hebbian and STDP (Figs 1H, 2A–2C, 3E, 3F, 4G and 4H). This observation indicated that the maximum firing rates during sleep, particularly in the Up state of SWO, are higher when the mean firing rates are equal in both sleep and wakefulness. The high maximum firing rates in sleep enhance synaptic learning rules, that is, Hebbian and STDP strengthen synaptic weights while Anti-Hebbian and Anti-STDP weaken synaptic weights. In vivo experiments also reported that the higher firing rates or shorter inter-spike interval (ISI) in the Up state of SWO [8,9,22] (S18 Fig). Higher maximum firing rates lead to greater $Ca^{2+}$ influx

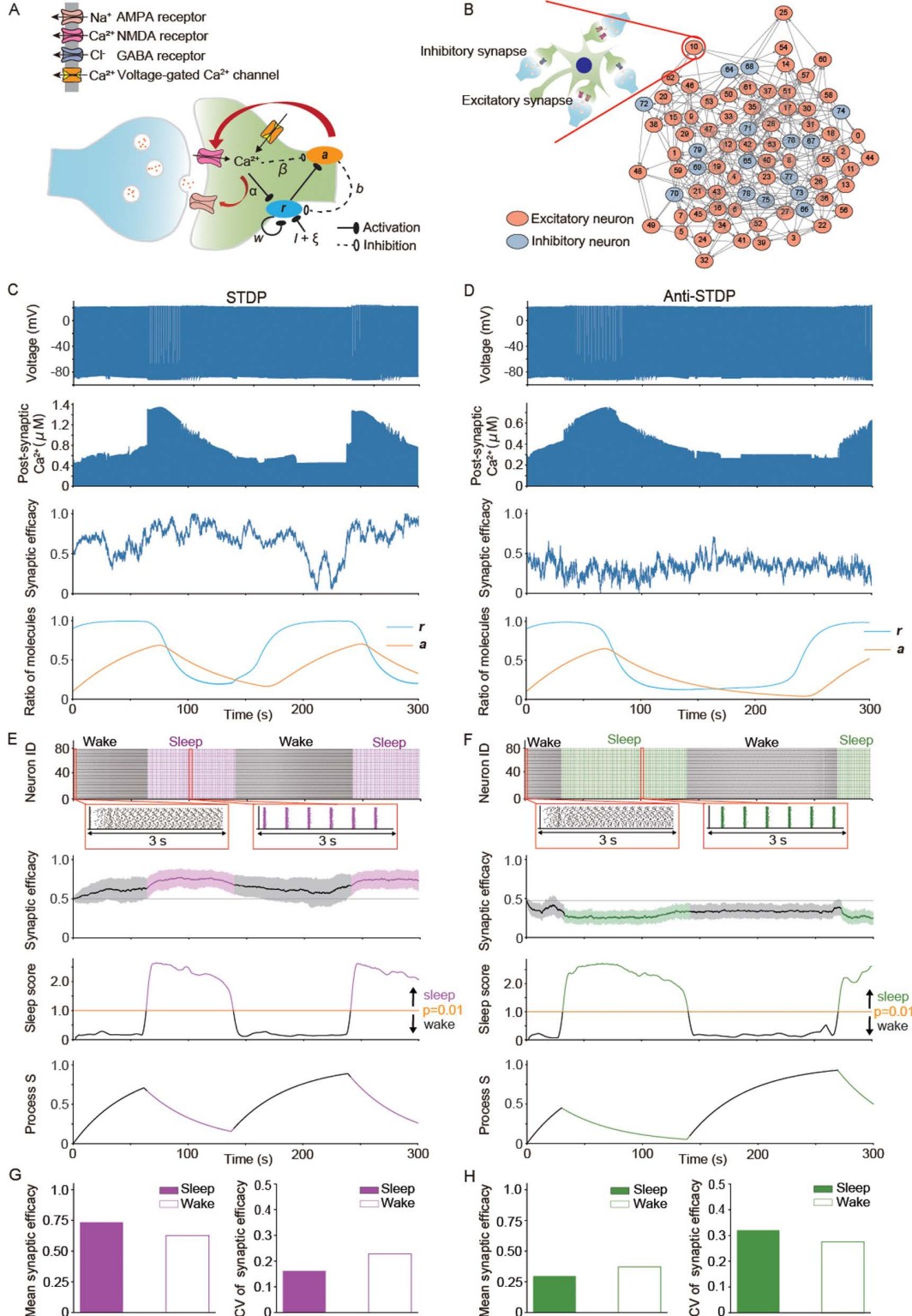

**Fig 4. WISE under Hebbian and STDP, and SHY under Anti-Hebbian and Anti-STDP is compatible with models including sleep–wake dynamics.** Synaptic efficacies were calculated in Hodgkin–Huxley-based network models bifurcated by the post-synaptic mechanism with sleep–wake dynamics under synaptic learning rules. The conductance of NMDAR was updated by ***a*** and the simulations were optimized by Pearson's correlation

coefficients between Process S and *r*. Initial synaptic efficacies of all synapses were 0.5. The simulations were started from a wake-like state and conducted for 500 s. The parameter set for channel or receptor conductances, learning rule and sleep–wake dynamics and initial values for variables in a representative model are shown in S5–S8 Tables. **(A)** Schematic illustration of the model for sleep–wake dynamics in excitatory synapses. The initial state of CaMKII, such as pT286/287 CaMKII is represented by *r*. The initial state *r* activates *a*, which corresponds to the second state of CaMKII, such as pT305/306 CaMKII, or phosphatases such as calcineurin. $\alpha$ and $\beta$ are coefficients for Ca$^{2+}$ activation for *r* and Ca$^{2+}$ inhibition for *a*, respectively. *w* is a coefficient for auto-activation of *r*. *b* is a coefficient for the inhibitory interaction of *a* against *r*. **(B)** Schematic illustration of the network model used in the simulations with sleep–wake dynamics. The network model has 80 neurons with the E:I ratio of 4:1. Each neuron has two excitatory and two inhibitory synapses. Connections between inhibitory neurons were not considered. **(C, D)** Time changes of post-synaptic membrane potential and Ca$^{2+}$ concentration, synaptic efficacy, and the ratio of two phosphorylated states of kinases (*r* and *a*) of a synapse in representative network models under STDP **(C)** and Anti-STDP **(D)**. The results of 0–300 s are shown. **(E, F)** Raster plots and time changes of mean synaptic efficacy, sleep score, and Process S in representative network models under STDP **(E)** and Anti-STDP **(F)**. The shadow in time changes in mean synaptic efficacy represents SD. The network was considered to be in the state of sleep or wake if the sleep score was above or below the threshold, respectively (the threshold is the value of sleep score where *p* = 0.01; see "Materials and methods, Evaluation of synchronization and desynchronization in Hodgkin–Huxley-based network models"). The results of 0–300 s are shown. **(G, H)** Mean and coefficient of variance (CV) of synaptic efficacy during the periods of sleep-like and wake-like states in representative network models under STDP **(E)** and Anti-STDP **(F)**. The data underlying the graphs shown in the figure can be found in Table F in S1 Data.

and larger changes in synaptic weights during SWO. Additionally, synchronization and hyperpolarized Down states that promote Ca$^{2+}$ influx in the subsequent Up state [39] likely contribute to elevated post-synaptic Ca$^{2+}$ during SWO.

We propose that synaptic changes depend on the differences in firing rates between NREM sleep and wakefulness. SHY may occur under STDP when mean firing rates during wakefulness are higher than during NREM sleep (Fig 5). A simulation showed that synaptic efficacies of neurons which are stimulated during wake-like states change according to SHY (see "Materials and methods, Calculation of synaptic efficacy under synaptic learning rules in Hodgkin–Huxley-based network models including stimulation during the wakefulness", and S15 Fig). Similarly, previous studies have demonstrated that exposure to novel stimuli or enforced wakefulness, conditions expected to increase sleep pressure, result in synaptic downscaling during sleep [40,41]. Conversely, quiet conditions, anticipated to yield lower sleep pressure during wakefulness, have strengthened synapses during sleep [40]. These observations align with our proposal because exposure to novel environments or higher activity increases firing rates during wakefulness, leading to SHY. In contrast, the quiet wake causes only limited differences in firing rate between NREM sleep and wakefulness, leading to WISE (Fig 5). Given that different brain regions become active at different levels during wakefulness or sleep deprivation, our results indicate that even opposite synaptic changes may occur across different brain regions. These different responses to the neuronal activities during wakefulness support the idea that NREM sleep normalizes the neuronal activities that are skewed during the wakefulness, as presented by Watson and colleagues [8].

Noteworthy, WISE predicts lower post-synaptic Ca$^{2+}$ concentration during wake-like desynchronized firing under Hebbian and STDP. This prediction aligns with the observation that calcineurin, an LTD-related molecule likely to be activated by lower Ca$^{2+}$ concentration during wakefulness, plays a role in excitatory post-neuronal synapses for generating SWO in the following NREM sleep [42–44]. Another prediction of WISE is synaptic connectivity homeostasis. When synaptic transmission is inhibited, the resulting SWO may strengthen synapses through WISE, compensating for the inhibited transmission. The connectivity homeostasis is also anticipated in the synaptic dynamics during hibernation. Decreases in firing rates and synaptic connections due to low temperatures during hibernation are associated with increases in SWO during NREM sleep and restoration of synaptic connections after hibernation [45,46], that is contrary to SHY. Since the lower the firing rates, the greater the synaptic potentiation during sleep in our results (Figs 1H and S9), WISE may explain synaptic dynamics during the hibernation. Likewise, in depressive disorder, which is characterized by reduced waking activity and dysfunction of AMPAR in frontal cortex [47], synaptic increase during NREM sleep may occur.

In conclusion, our study provides a unified framework for the synaptic dynamics during the sleep–wake cycle (Fig 6). It suggests that SHY, WISE, and the normalization during NREM sleep coexist but occur depending on synaptic learning rules and neuronal activities of networks. Although further studies are needed to investigate relationships between

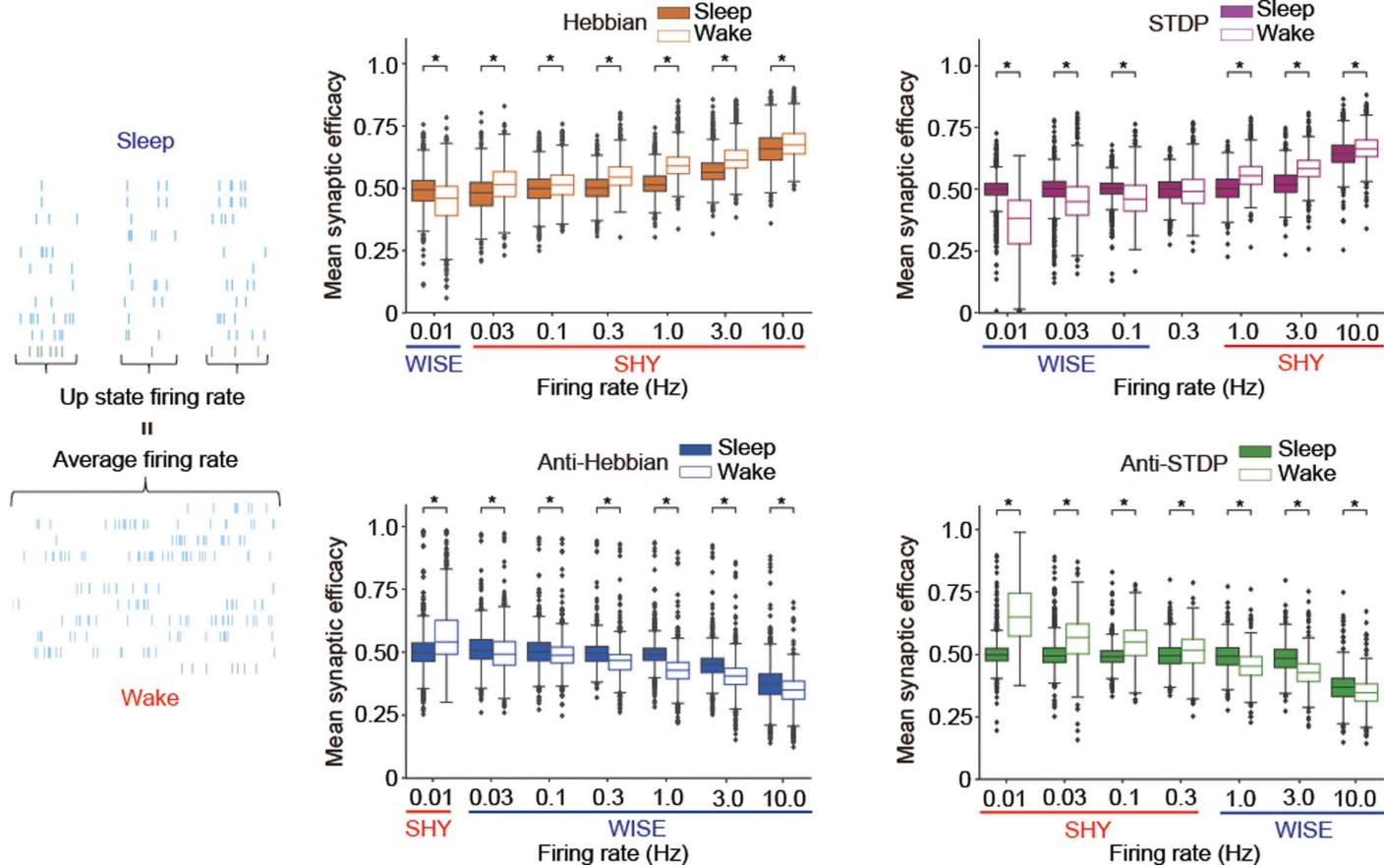

**Fig 5. Synaptic changes depend on firing rates assuming higher firing rates during wakefulness.** A Schematic illustration for sleep-like and wake-like spike patterns and box plots for mean synaptic efficacy in sleep-like and wake-like firing patterns for different learning rules and mean firing rates ($n$ = 1,000 for each firing rate, $n$ represents the number of synaptic learning rules). The mean firing rates in wake-like states are equal to the mean firing rates in Up states of sleep-like states. The sleep-like firing patterns were generated by sampling from the lognormal distributions with $\log_{10}$(mean Up-state duration) = 2.7 and $\log_{10}$(mean Down-state duration) = 3.0. SD was calculated according to the linear regression analysis based on in vivo data (see "Materials and methods", S18 Fig). All the neurons were represented by the equation (Eq. 1–10). A total of 1,000 parameter sets for the specific learning rules collected by parameter search in Fig 1 were applied. Initial synaptic efficacies in all the synapses were 0.5, and synaptic efficacies were simulated for 6 min. Synaptic efficacies for the last 2 min were averaged. Mean firing rates of the wake-like states are shown in the x-axis. WISE and SHY dynamics are highlighted under the x-axis according to the change in the direction of synaptic efficacy. The whiskers above and below show minimal to maximal values. The box extends from the 25th to the 75th percentile and the middle line indicates the median. Bayesian statistical analysis was performed using Markov Chain Monte Carlo method to infer posterior distributions of average differences in mean synaptic efficacy between sleep-like firing patterns and wake-like firing patterns. Asterisks (*) indicate 95% CIs do not include zero. The data underlying the graphs shown in the figure can be found in Table G in S1 Data. The 95% CIs for the distributions of average differences are shown in Table F in S3 Data.

synaptic learning rules, neuronal activities, and synaptic weights, the framework we presented here lays a foundation for future research.

## Synaptic dynamics and brain functions

Our study provides several implications regarding the relationship between synaptic dynamics and brain functions during the sleep–wake cycle. WISE supports the notion that synaptic potentiation in NREM sleep contribute to memory consolidation [4,38], which can be promoted by STDP [48,49]. On the other hand, we revealed that wake-like desynchronized states can lead to synaptic depression under Hebbian and STDP, especially at lower firing rates (Figs 1H, 2A–2C, 3E,

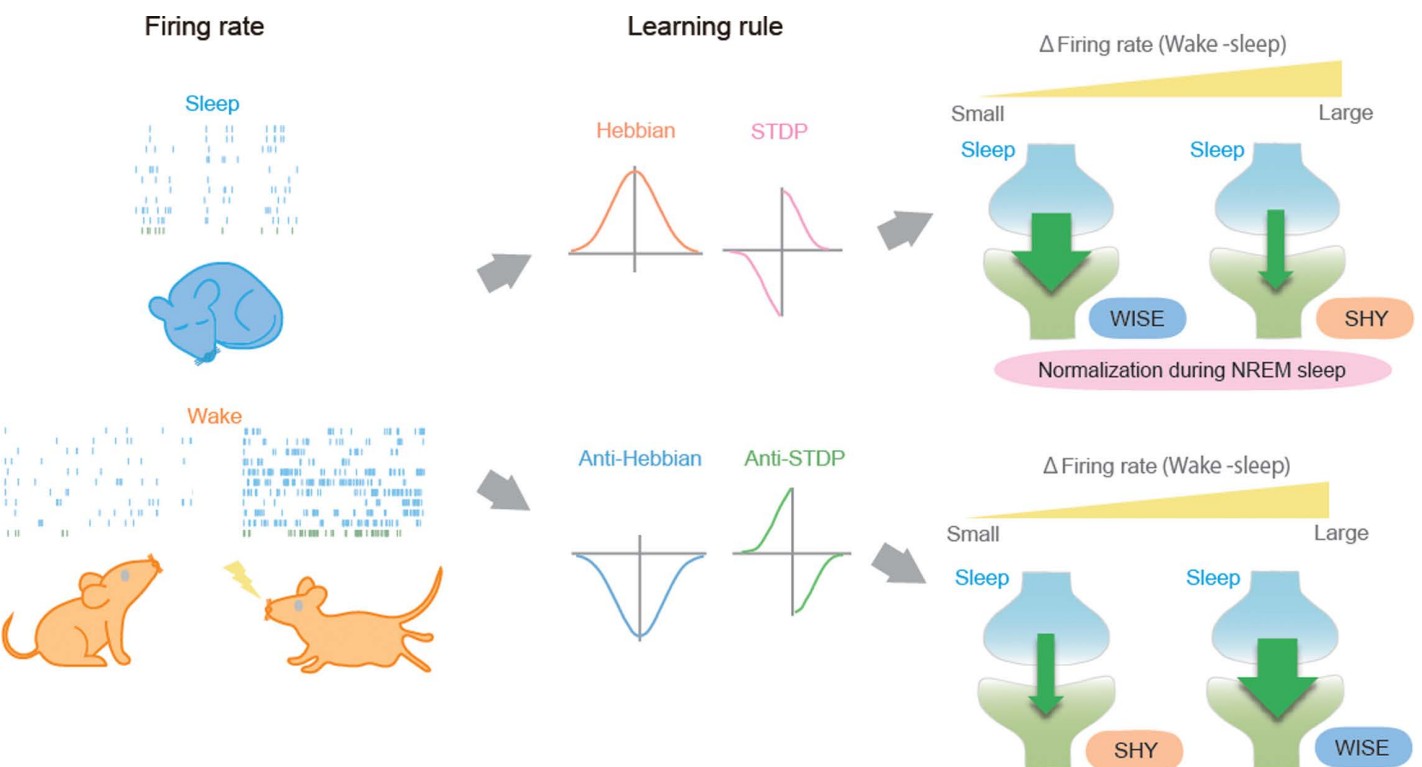

**Fig 6. A unified framework for synaptic dynamics during the sleep–wake cycle.** A Graphical abstract of a unified framework for synaptic dynamics during the sleep–wake cycle. SHY: synaptic homeostasis hypothesis, WISE: wake inhibition sleep excitation.

3F, 4G and 4H). One implication of this inhibition during desynchronized states is the enhancement of the signal to noise ratio (SNR) [50,51]. Stimulated neurons likely fire at higher rates, and their synaptic efficacies become larger than those of other neurons in the background desynchronized states. Additionally, neurotransmitter release during wakefulness possibly modulates learning rules to control SNR [52]. Indeed, it has been reported that cholinergic projections from the nucleus basalis to the cortex are responsible for synaptic inhibition [51].

We also found that the variability of synaptic efficacy is higher during wakefulness than in NREM sleep under Hebbian and STDP (Figs 4G, S12–S14 and S16). Several studies suggest that the variability of synaptic efficacies leads to the variability of network activity and reflects probabilistic inference for external worlds and decision-making [53–55]. In this context, our results suggest that awake states, with higher variability of synapses, are advantageous for exploration and behavioral selection. Additionally, the decline in synaptic variability during NREM sleep under Hebbian and STDP supports the idea of normalization of firing rates, whose distribution is skewed during wakefulness [8].

## Model characteristics and limitations

Some computational studies have investigated synaptic changes in neural networks under STDP protocols using $Ca^{2+}$-based plasticity models [21,56] while other studies have also examined how SWO affect synaptic plasticity under STDP conditions [49]. However, these previous studies were limited to a single synaptic learning rule or firing pattern. Our study is the first to comprehensively investigate synaptic dynamics during sleep–wake cycle by integrating $Ca^{2+}$-based plasticity model to represent various types of synaptic learning rules and various simulated sleep–wake firing patterns. By selecting model parameters that align with experimentally observed firing patterns and synaptic learning rules, our

model can predict synaptic dynamics under specific conditions regardless of species, brain region, or sex. In the simple model (where firing patterns are given and do not change under simulation) used in Figs 1H, 2, and 5, sleep-like or wake-like firing patterns were generated based on the distributions of ISI, Up-state duration, and Down-state duration estimated from experimentally observed rat firing patterns [8,22]. Synaptic efficacy became higher at higher mean firing rates in both states when adjusting mean firing rates by changing ISI (Fig 5) while synaptic efficacy during sleep-like firing patterns seems to be independent when adjusting mean firing rates by changing Down-state durations as shown in Fig 1H. Regardless of ways to generate spike patterns during sleep-like states, synaptic efficacy during sleep tends to become higher than during wakefulness, assuming the same mean firing rates between both states (Fig 2D and 2E). The results for other species can be obtained by simulating firing patterns using the same method (detailed in "Materials and methods, Generation of sleep and wake-like firing patterns") when these distributions are known. The ranges of Up- and Down-state durations during SWO in mice, rats, and cats are approximately 100–500 ms [9,57], while in humans, Up-state durations range from 250–1,000 ms [58], all of which fall within the ranges examined in Figs 2D and 2E. Similarly, wake-state ISI across various species typically range from 2 to 100 ms [9,59], mostly within the scope covered in Fig 2E. Therefore, we suppose our finding in the present study captured universal aspects of synaptic dynamic in the sleep and wake cycles regardless of species, brain region, or sex.

Synaptic homeostasis is a type of more long-term plasticity that returns a neuron back to its homeostatic set point [60]. In our models, we did not explicitly include synaptic homeostasis in the preposition but consider synaptic homeostasis in the definitions of SHY and WISE. For example, we assume that SHY upscales synaptic strength during wakefulness and downscales during sleep to achieve synaptic homeostasis while WISE upscales synaptic strength during sleep and downscales during wakefulness to achieve synaptic homeostasis. Importantly, since both SHY and WISE can achieve synaptic homeostasis, adding the further mechanism of synaptic homeostasis in the preposition would not alter our predictions.

The present study has some limitations. Our models are composed of uniform neuronal populations and synaptic learning rules, which can be regulated by sleep promoting kinases. However, different types of neurons are interconnected in the cortex, and their firing rates and synaptic learning rules vary by region and condition [11,52,61,62]. Further studies on synaptic plasticity across a wide variety of neurons, regions, and conditions will provide a more detailed understanding of the relationship between synaptic learning rules and synaptic plasticity. In this study, we assumed that the activation of sleep-promoting kinases and the accumulation of phosphorylation during wakefulness induce SWO. We presented three bifurcation models for the generation of SWO based on the changes in firing patterns (Figs 3 and 4). Since the molecular mechanism of SWO is still under vigorous investigations, these bifurcation models will need to be updated by future experiments. Additionally, the model for the sleep–wake dynamics has some ambiguities (Fig 4). The coefficient "b" in the equation (Eq. 14, "Materials and methods") can represent not only the second phosphorylated states of CaMKII such as pCaMKII (T305/T306), but also phosphatases such as calcineurin [42]. The homeostatic oscillation caused by CaMKII described in Fig 4 is one of the examples. We consider that our model is not limited to CaMKII, and salt-inducible kinase 3 (SIK3) and extracellular signal-regulated kinase (ERK) are alternative candidates because these kinases also have multiple phosphorylation states and are related to circadian rhythm [63–68]. Moreover, it is unclear which phosphorylation state of CaMKII highly correlates with Process S or interacts with the molecules that induce SWO. Our model incorporated two principal phosphorylation states of CaMKII—designated as "first state" and "second state"—both potentially interacting with molecules to generate SWO and exhibiting positive correlations with Process S. In S11A Fig, we systematically conducted simulations with varying parameters across potential combinations. We selected one where a specific phosphorylation state exhibited maximal Pearson's correlation coefficients with Process S. The results suggest that the first state (expecting CaMKII pT286/T287) demonstrates a stronger correlation with Process S, while the second state appears more intimately involved in generation of the SWO. These computational predictions require experimental validation in subsequent studies.

The present computational study also does not explicitly incorporate the effects of firing patterns on neuropeptide or hormone releases. Previous studies demonstrated that burst firing promotes the release of neuropeptides such as Substance P and Neuropeptide Y, which significantly contribute to modulation of neural circuits [69,70]. Although systematic investigations of the relationship between firing patterns and neurotransmitter release in cortical neurons remain limited, several studies have documented the release of neuropeptides, including Neuropeptide Y, somatostatin (SST), and vasoactive intestinal peptide (VIP) alter the thresholds in STDP in cortex [71]. In contrast with those of classical neurotransmitters, these neuropeptides predominantly exert their effects through G-protein coupled receptors, functioning over extended temporal domains, potentially introducing additional complexity to plasticity mechanisms [72]. Elucidating the contributions of neuropeptides and hormones influenced by sleep–wake firing patterns to synaptic plasticity and homeostatic control of neural excitability in cortical neurons represents an important direction for subsequent studies.

## Materials and methods

### Modeling synaptic learning rules

Synaptic efficacy variable $\rho$ is described by a first order differential equation according to the previous article [21].

$$\tau_s \frac{d\rho}{dt} = -\rho(1-\rho)(\rho^*-\rho) + \gamma_p(1-\rho)\Theta[c-\theta_p] - \gamma_d\rho\Theta[c-\theta_d] + Noise \tag{1}$$

($\Theta$ denotes the Heaviside function: $\Theta[c-\theta] = 0$ for $c < \theta$ and $\Theta[c-\theta] = 1$ for $c \geq \theta$).

$c$ is the post-synaptic $Ca^{2+}$ concentration and expressed as the sum of $Ca^{2+}$ influx from NMDAR ($C_{pre}$) and VGCC ($C_{post}$).

$$c = C_{pre} + C_{post} \tag{2}$$

$C_{pre}$ follows a second-order differential equation of NMDAR activation by firing of pre-synaptic neurons. We modified the NMDAR kinetics (using smaller time constant of $s_{NMDA}$ and $x_{NMDA}$ than that in the original model) and did not consider $Mg^{2+}$ block against NMDAR for the purpose of obtaining parameter sets for the four different types of synaptic learning rules.

$$\frac{dC_{pre}}{dt} = -\alpha_{Ca}I_{Ca\_NMDA}\beta_{NMDA} - \frac{C_{pre}}{\tau_{pre}} \tag{3}$$

$$I_{Ca\_NMDA} = g_{NMDA}s_{NMDA}(V_{rest} - V_{Ca}) \tag{4}$$

$$\frac{ds_{NMDA}}{dt} = a_{sNMDA}x_{NMDA}(1 - s_{NMDA}) - \frac{s_{NMDA}}{\tau_{sNMDA}} \tag{5}$$

$$\frac{dx_{NMDA}}{dt} = a_{xNMDA}f(V_{pre}) - \frac{x_{NMDA}}{\tau_{xNMDA}} \tag{6}$$

$$f(V) = 1/[1 + exp(-(V-20)/2)] \tag{7}$$

$C_{post}$ is described by a first order differential equation of VGCC

$$\frac{dC_{post}}{dt} = -\alpha_{Ca}AI_{Ca\_VGCC}\beta_{VGCC} - \frac{C_{post}}{\tau_{post}} \tag{8}$$

$$I_{Ca\_VGCC} = g_{Ca}m^2{}_{Ca\infty}(V_{post})(V_{post} - V_{Ca}) \tag{9}$$

$$m_{Ca\infty}(V) = 1/[1 + exp(-(V+20)/9) \tag{10}$$

Equations (Eq. 3–10) were based on the previous study [23,73,74]. $I_{Ca\_VGCC}$ and $I_{Ca\_NMDA}$ are Ca²⁺ current through NMDAR and VGCC, respectively. $x_{NMDA}$ and $s_{NMDA}$ are gating variables for second-order kinetics. $f$ is a saturating function of membrane potential $V$ ($V_{pre}$ and $V_{post}$ are membrane potentials in pre-synaptic and post-synaptic neurons, respectively) and $m_{Ca\infty}(V)$ is a steady-state activation gating variable which is dependent on $V$.

Fixed parameters are shown in S1 Table. Scaling parameters $\beta_{NMDA}$ and $\beta_{VGCC}$ in equations (Eq. 3 and 8) were estimated so that the mean amplitude of $C_{pre}$ = 0.7 $\mu M$ and the mean amplitude of $C_{post}$ = 1.4 $\mu M$ according to the results of the previous studies [75]. The equation for noise and the comparison of simulation and analytical results are in "Materials and methods, Comparison of simulation and analytical results for synaptic learning rules" and S17 Fig.

**Parameter search for synaptic learning rules**

We selected parameter values from uniform distributions within the ranges of values [21] (S2 Table). Parameter sets for Hebbian and STDP were collected in the conditions where $\theta_p > \theta_d$ because most of them did not appear in $\theta_p < \theta_d$. In the same way, parameter sets for Anti-Hebbian and Anti-STDP were collected in $\theta_p < \theta_d$. The synaptic changes in different time lags were calculated analytically according to the way presented by previous reports [21]. Neurons were stimulated at 1 Hz for 60 s. First, scaling parameters $\beta_{NMDA}$ and $\beta_{VGCC}$ in the equations (Eq. 3 and 8) were calculated so that the mean amplitudes of $C_{pre}$ and $C_{post}$ ($MC_{pre}$ and $MC_{post}$) during 2–3 s satisfy $MC_{pre}$ = 0.7 $\mu M$ and $MC_{post}$ = 1.4 $\mu M$ because this kind of relationship holds in the experimental studies [75]. We considered that synaptic weights did not change if spike-pairs had large time differences. Specifically, the ratio of $\gamma_p$ and $\gamma_d$ was calculated and the value of $\gamma_d$ was updated so that the potentiation and depression rate become equal when the lag time was equal to 100 ms. The potentiation and depression rate were expressed as $\gamma_p\alpha_p$ and $\gamma_d\alpha_d$, respectively ($\alpha_x = \frac{1}{T}\int_0^T \Theta[c(t) - \theta_x]dt$, $T$ is the duration of the stimulation protocol) [21]. Then, the scaling parameters were calculated again in the parameter sets where $\gamma_d$ was updated and synaptic changes were analytically calculated in each lag time (from −160 ms to 160 ms with 10 ms intervals). To speed up the calculations, potentiation and depression rate were calculated during the 2–3 s and multiplied by 60. Gaussian curves that represent four different types of synaptic learning rules were described by the following equations with amplitude ($a_{lr}$, $b_{lr}$, $|a_{lr}| = |b_{lr}|$) and time constant ($\tau_{lr}$).

$$a_{lr}\exp\left(-\left(\frac{x}{\tau_{lr}}\right)^2\right) \quad (x \geq 0)$$

$$b_{lr}\exp\left(-\left(\frac{x}{\tau_{lr}}\right)^2\right) \quad (x < 0)$$

For each parameter set, the sum of squared errors (SSE) between analytical solutions and fitting gaussian curves were calculated and the parameter set was collected if the SSE was less than the threshold (The threshold was 0.45 in $a_{lr}$ = 0.9, $\tau_{lr}$ = 50 or 0.25 in other conditions of Figs 1E and S1A–S1E. The threshold was 0.6 when searching for synaptic learning rules in Hodgkin–Huxley-based network models).

**Noise term and comparison of simulation and analytical results**

The noise term in the equation (Eq. 1) was expressed by the following equation.

$$Noise\,(t) = \sigma\sqrt{\frac{\pi\,[\Theta\,(c(t) - \theta_d) + \Theta\,(c(t) - \theta_p)]\,z}{dt}}\,\eta(t)$$

When solving the differential equations, the noise term was calculated by above equation and added to the differential equation. $z$ is the coefficient of noise, $dt$ is a step size, and $\eta(t)$ is a Gaussian white noise process with unit variance

density. We simulated the synaptic efficacy by the equation (Eq. 1) for 1,000 times and calculated the synaptic changes in different lag times according to the previous article [21]. An analytical solution was compared to simulation results in different values of $z$ in the simple model (S17 Fig) and $z = 3.5$ was used in the simulations with synaptic learning rules in this study.

### Generation of sleep and wake-like spike patterns

We assumed that synchronization of firing was observed in NREM sleep states and desynchronization of firing was observed during wakefulness. Timestamps of wake-like spikes were obtained from ISI per spike, and those of sleep-like spikes were obtained from ISI per spike and duration of Up and Down states per burst. ISI per spike and durations of Up and Down states per burst were sampled from lognormal distributions with specific means and SDs. Spike patterns with a certain mean firing rate in Figs 1H, 2A–2D, S3, S16A, and S19 were generated by adjusting the mean Down-state duration. Spike patterns in Figs 2E, 5, and S2 were generated by adjusting the mean ISI. The relationships between mean and SD for the lognormal distributions were obtained based on previous in vivo recordings [8,22] (S18C–S18G Fig).

### Definition of lognormal distributions based on in vivo recordings

The duration of Up states and Down states were sampled from the lognormal distributions because previous studies found that duration of Up states and Down states have lognormal distributions [8,76]. We assumed that ISI also has lognormal distributions on the same datasets [8,22] (S18A and S18B Fig). The spike data recorded from excitatory neurons (verified by cross-correlogram) of rats which had natural sleep–wake cycle from the previous studies [8,22] were used in this study. To reduce the variables, liner regression analysis was performed on the ordinary logarithm of mean and SD of the duration of Up and Down states (S18C and S18D Fig) and ISI (S18E and S18F Fig), and the ordinary logarithm of the mean Up-state duration and mean Down-state durations (S18G Fig). Then, the SD of Up-state duration, Down-state duration, ISI in the Up states of sleep-like firing patterns and ISI in wake-like firing patterns were calculated by the equations: $0.35 \times \log_{10}(\textit{mean Up-state duration}) - 0.7$, $0.25 \times \log_{10}(\textit{mean Down-state duration}) - 0.35$, $-0.2 \times \log_{10}(\textit{mean ISI in the Up states of sleep}) + 0.95$, and $0.03 \times \log_{10}(\textit{mean ISI in the state of wake}) + 0.65$, respectively. The mean Down-state duration was calculated by the equation $-0.7 \times \log_{10}(\textit{mean Up-state duration}) + 4.0$.

### Conversion from spike patterns to voltage waveforms

The timestamp data for spikes during sleep-like and wake-like firing patterns were converted to voltage waveforms. We defined several parameters when constructing voltage waveforms so that the timestamps of the spikes corresponded to the peaks of the membrane potential by referring to the experimental studies [9,57] (S3 Table). Voltage waveforms were generated by linear interpolation of the points for the membrane potential of peaks, after hyperpolarization and Up states. The difference of membrane potentials between Up states and Down states was 15 mV when calculating synaptic efficacy in Figs 1H, 2A–2E, 5, S2, S3 and S16. The results in other values for the difference of membrane potentials are shown in S19 Fig.

### Comparison of synaptic efficacy during sleep-like and wake-like firing patterns in simple model

The generated sleep-like or wake-like spike trains were converted to the data for membrane potential (see "Materials and methods, Conversion from spike patterns to voltage waveforms"). Time changes in synaptic efficacy were calculated by assigning the time-series data for sleep-like and wake-like membrane potentials and parameter sets for synaptic learning rules to the equations (Eq. 1–10). Time changes in synaptic efficacy were calculated for 6, 18, and 36 min with mean firing rates of pre-synaptic neurons being 0.1–10.0 Hz, 0.03 Hz and 0.01 Hz, respectively. Mean and CV of synaptic efficacy for the last two minutes were calculated by every time step and compared between sleep-like and wake-like firing patterns as an average over time. In random connections of Fig 2B, each neuron was randomly connected to other neurons with a probability of 12% and the synaptic efficacy was calculated based on their spike trains.

## Hodgkin–Huxley-based network model

A Hodgkin–Huxley-based network model was constructed based on the averaged neuron model in a previous study with some modifications as follows [23,73,74]:

$$CA\frac{dV}{dt} = -A\left(I_L\left(V_{post}\right) + I_{Na}\left(V_{post},\ h_{na}\right) + I_K\left(V_{post},\ n_k\right) + I_A\left(V_{post},\ h_A\right) + I_{KS}\left(V_{post},\ m_{KS}\right)\right.$$
$$\left. + I_{Ca}\left(V_{post}\right) + I_{KCa}\left(V_{post},\ \left[Ca^{2+}\right]\right) + I_{NaP}\left(V_{post}\right) + I_{AR}\left(V_{post}\right)\right)$$
$$- I_{NMDA}\left(V_{post},\ s_{NMDA}, x_{NMDA}\right) - I_{AMPA}\left(V_{post},\ s_{AMPA}\right) - I_{GABA}\left(V_{post},\ s_{GABA}\right)$$

where $C$ is the membrane capacitance, $A$ is the area of a single neuron, and $V$ is the membrane potential. The intrinsic (non-synaptic) ion channels in the cell body were modeled using Hodgkin–Huxley-based equations, with a gating variable x governed by the first-order kinetics equation:

$$\frac{dx}{dt} = \varphi\left[\alpha_x\left(V_{post}\right)\left(1-x\right) - \beta_x\left(V_{post}\right)x\right] = \varphi\left[\ x_\infty(V) - x\right]/\tau_x(V)$$

where $\varphi$ is the temperature factor ($\varphi = 1$).
Intrinsic currents are given by the following equations:

$$I_L\left(V_{post}\right) = g_L\left(V_{post} - V_L\right)$$

$$I_{Na}\left(V_{post},\ h_{na}\right) = g_{Na}m^3_{Na\infty}\left(V_{post}\right)h_{Na}\left(V_{post} - V_{Na}\right)$$

$$m_{Na\infty}\left(V_{post}\right) = \alpha_m(V_{post})/\left(\alpha_m\left(V_{post}\right) + \beta_m\left(V_{post}\right)\right)$$

$$\alpha_m\left(V_{post}\right) = 0.1(V_{post} + 33)/\left[1 - exp\left(-\left(V_{post} + 33\right)/10\right)\right]$$

$$\beta_m\left(V_{post}\right) = 4exp(-\left(V_{post} + 53.7\right)/12)$$

$$\frac{dh_{na}}{dt} = 4(\alpha_h\left(V_{post}\right)\left(1 - h_{Na}\right) - \beta_h\left(V_{post}\right)h_{Na})$$

$$\alpha_h\left(V_{post}\right) = 0.07exp(-V_{post} + 50)/10)$$

$$\beta_h\left(V_{post}\right) = 1/\left[1 + exp(-(V_{post} + 20)/10)\right]$$

$$I_K\left(V_{post},\ n_k\right) = g_K n^4_K(V_{post} - V_K)$$

$$\frac{dn_k}{dt} = 4(\alpha_n\left(V_{post}\right)\left(1 - n_k\right) - \beta_n\left(V_{post}\right)n_k)$$

$$\alpha_n\left(V_{post}\right) = 0.01(V_{post} + 34)/[1 - exp\left(-\left(V_{post} + 34\right)/10\right)]$$

$$\beta_n\left(V_{post}\right) = 0.125exp(-\left(V_{post} + 44\right)/25)$$

$$I_A\left(V_{post},\ h_A\right) = g_A m_{A\infty}^3(V_{post})h_A(V_{post} - V_K)$$

$$m_{A\infty}\left(V_{post}\right) = 1/[1 + exp(-(V_{post} + 50)/2)]$$

$$\frac{dh_A}{dt} = (h_{A\infty}\left(V_{post}\right) - h_A)/\tau_{hA}$$

$$h_{A\infty}\left(V_{post}\right) = 1/[1 + exp((V_{post} + 80)/6)]$$

$$I_{KS}\left(V_{post},\ m_{KS}\right) = g_{KS}m_{KS}(V_{post} - V_K)$$

$$\frac{dm_{KS}}{dt} = (m_{KS\infty}\left(V_{post}\right) - m_{KS})/\tau_{mKS}\left(V_{post}\right)$$

$$m_{KS\infty}\left(V_{post}\right) = 1/[1 + exp(-(V_{post} + 34)/6.5)]$$

$$\tau_{mKS}\left(V_{post}\right) = 8/[exp(-(V_{post} + 55)/30) + exp((V_{post} + 55)/30)]$$

$$I_{Ca}\left(V_{post}\right) = g_{Ca}m_{Ca\infty}^2(V)(V_{post} - V_{Ca})$$

$$m_{Ca\infty}\left(V_{post}\right) = 1/[1 + exp(-(V_{post} + 20)/9)]$$

$$I_{KCa}\left(V_{post},\ \left[Ca^{2+}\right]\right) = g_{KCa}m_{KCa\infty}(\left[Ca^{2+}\right])(V_{post} - V_K)$$

$$m_{KCa\infty}\left(\left[Ca^{2+}\right]\right) = 1/[1 + (K_D/[Ca^{2+}])^{3.5}]$$

$$I_{NaP}\left(V_{post}\right) = g_{NaP}m_{NaP\infty}^3(V_{post})(V_{post} - V_{Na})$$

$$m_{NaP\infty}\left(V_{post}\right) = 1/[1 + exp(-(V_{post} + 55.7)/7.7)]$$

$$I_{AR}\left(V_{post}\right) = g_{AR}h_{AR\infty}(V_{post})(V_{post} - V_K)$$

$$h_{AR\infty}\left(V_{post}\right) = 1/[1 + exp((V_{post} + 75)/4)]$$

where $n_k$, $h_{Na}$, $h_A$, and $m_{KS}$ are gating variables and $m_{Na\infty}$, $m_{A\infty}$, $h_{A\infty}$, $m_{KS\infty}$, $m_{Ca\infty}$, $m_{KCa\infty}$, $m_{NaP\infty}$, and $h_{AR\infty}$ are steady-state activation gating variables, and $\alpha_m$, $\beta_m$, $\alpha_h$, $\beta_h$, $\alpha_n$ and $\beta_n$ are rate constants.

The extrinsic (synaptic) ion currents in the dendrite were also modeled using Hodgkin–Huxley-based equations, with a gating variable $s$ governed by the first-order kinetics equation for AMPA and GABA receptors:

$$\frac{ds}{dt} = a_s f(V) - \frac{s}{\tau_s}$$

and with gating variables $s$ and $x$ governed by the second-order kinetics equations for NMDA receptors:

$$\frac{ds}{dt} = a_s x(1-s) - \frac{s}{\tau_s}$$

$$\frac{dx}{dt} = a_x f(V) - \frac{x}{\tau_x}$$

where $\tau$ is the time constant for turnover of $s$ and $x$, and $f$ is a saturating function of $V$:

$$f(V_{post}) = 1/[1 + exp(-(V_{post} - 20)/2)]$$

Extrinsic currents are then given by the following equations:

$$I_{AMPA}(V_{post}, s_{AMPA}) = g_{AMPA} s_{AMPA}(V_{post} - V_{AMPA})$$

$$\frac{ds_{AMPA}}{dt} = a_{AMPA} f(V_{pre}) - \frac{s_{AMPA}}{\tau_{AMPA}}$$

$$I_{NMDA}(V_{post}, s_{NMDA}, x_{NMDA}) = g_{NMDA} s_{NMDA}(V_{post} - V_{NMDA})$$

$$\frac{ds_{NMDA}}{dt} = a_{sNMDA} x_{NMDA}(1 - s_{NMDA}) - \frac{s_{NMDA}}{\tau_{sNMDA}}$$

$$\frac{dx_{NMDA}}{dt} = a_{xNMDA} f(V_{pre}) - \frac{x_{NMDA}}{\tau_{xNMDA}}$$

$$I_{GABA}(V_{post}, s_{GABA}) = g_{GABA} s_{GABA}(V_{post} - V_{GABA})$$

$$\frac{ds_{GABA}}{dt} = a_{GABA} f(V_{pre}) - \frac{s_{GABA}}{\tau_{GABA}}$$

The Ca²⁺ concentration in cell body and synapses were described by the following equations:

$$\frac{d[Ca^{2+}]}{dt} = -\alpha_{Ca}(AI_{Ca}(V_{post})) - \frac{\left[Ca^{2+}\right]}{\tau_{Ca}}$$

$$\frac{d[Ca^{2+}_{syn}]}{dt} = -\alpha_{Ca}(AI_{Ca}(V_{post})\beta_{VGCC} + I_{NMDA}(V_{post}, s_{NMDA}, x_{NMDA})\beta_{NMDA}) - \frac{\left[Ca^{2+}_{syn}\right]}{\tau_{Ca-syn}}$$

The constant values are listed in S4 Table. Intrinsic (non-synaptic) ion currents (e.g., $I_{Na}$) should be multiplied by 10 to adjust its unit to nanoampere (nA) when the numerical values listed in S5 Table are directly used in the numerical simulations.

 

The network model has two $Ca^{2+}$ compartments of cell body and synapse following the facts that the time constant of $Ca^{2+}$ in synapses is very short (about 15 ms) [75] while a larger $Ca^{2+}$ time constant (about 50–1,000 ms) is necessary to induce SWO [23]. We did not consider the influx of $Ca^{2+}$ from spines to a cell body taking into account the immediate uptake of $Ca^{2+}$ by intracellular buffers [77]. The number of excitatory and inhibitory synapses per neuron were sampled from lognormal distributions, respectively. Connections between inhibitory neurons were not considered because we did not include this type of connection in parameter search and bifurcation analysis in a single neuron. The mean = 2 and SD = 0.01 was adopted for lognormal distributions of the number of excitatory and inhibitory synapses per neuron in Figs 3 and 4, in which excitatory neurons had both two excitatory and inhibitory synapses and inhibitory neurons had only two excitatory synapses. The conductance of AMPAR and NMDAR or the conductance of GABAR were divided by the average number of excitatory or inhibitory synapses per neuron respectively in the simulations of Hodgkin–Huxley-based network models.

## Parameter search for SWO and bifurcation analysis in Hodgkin–Huxley-based network models

Parameter search for SWO and bifurcation analysis for a single neuron were based on the previous article [23]. First, we randomly generated parameter sets from a large parameter space and searched for parameter sets that yield firing patterns of SWO. The ranges of parameters were defined to include biophysically reasonable values. The conductance of intrinsic (non-synaptic) channels in the soma and axon ($g_L$, $g_{Na}$, $g_K$, $g_A$, $g_{KS}$, $g_{Ca}$, $g_{KCa}$, $g_{NaP}$, $g_{AR}$) and extrinsic (synaptic) channels in the dendrite ($g_{AMPA}$, $g_{NMDA}$, $g_{GABA}$) were generated by selecting parameters from an exponential distribution bounded to the interval $10^{-2}$–$10^2$ mS/cm$^2$ and $10^{-3}$–$10^1$ µS, respectively. A random parameter search for time constant of intracellular $Ca^{2+}$ ($\tau_{Ca}$) were also conducted in the interval $10^1$–$10^3$ ms. The parameter sets that yielded a steady-state and periodic solution with real values were evaluated. The major frequency of the oscillatory behavior was analyzed by fast Fourier transform. Additionally, we assessed the detailed structure of the oscillations by counting the number of spikes; the number of spikes was determined as half the number of times the membrane potential crossed −20 mV. Solutions in which the maximum of membrane potential were more than 200 mV or the minimum of membrane potential were less than −200 mV were eliminated at this point. Based on these characteristics, the solutions were classified into six categories: "Resting" (spike numbers per second < 0.6, peak frequency ≤ 0.5 Hz, maximum membrane potential < −30 mV or minimum membrane potential > −30 mV), "SWO" (peak frequency ≥ 0.5 Hz and 2 × peak frequency ≤ spike number per second < 30), "SWO with high frequency" (peak frequency ≥ 0.5 Hz, spike number per second ≥ 2 × peak frequency and spike number per second ≥ 30), "AWAKE" (peak frequency ≥ 0.5 Hz, spike number per second ≤ 2 × peak frequency and spike number per second < 30), "AWAKE with high frequency" (peak frequency ≥ 0.5 Hz, 30 < spike number per second ≤ 2 × peak frequency and spike number per second < 100) and "EXCLUDED" (others not classified above).

Second, we conducted bifurcation analysis in parameter sets that yield SWO or SWO with high frequency. In the bifurcation analysis, channel or receptor conductance was gradually changed from $10^{-2}$ to $10^2$ times its original value by 100 steps (for the bifurcation by the pre-synaptic mechanism, $a_{AMPA}$, $a_{xNMDA}$, $a_{GABA}$ were gradually changed from $10^{-2}$ to $10^1$ times its original value by 100 steps), and the solutions were classified into six categories according to the criteria described above. We selected parameter sets that including the bifurcation from "AWAKE" to "SWO" and satisfying minimum membrane potential of "SWO" + 5 mV < minimum membrane potential of "AWAKE" in continuously varying conductance. In the parameter search and bifurcation analysis for a single neuron, the simulations were conducted for 6 s, and the results of 1–6 s were analyzed. Integration was performed with the following initial values in each simulation of parameter search and bifurcation analysis for a single neuron: $V$ = −45 mV, $h_{Na}$ = 0.045 (unitless), $n_K$ = 0.54 (unitless), $h_A$ = 0.045 (unitless), $m_{KS}$ = 0.34 (unitless), $[Ca^{2+}]$ = 1 µM, $s_{AMPA}$ = 0.01 (unitless), $s_{NMDA}$ = 0.01 (unitless), $x_{NMDA}$ = 0.01 (unitless), $s_{GABA}$ = 0.01 (unitless).

Then, the network model of 80 neurons was constructed with E:I ratio of 4:1 and bifurcation analysis for network models was conducted. In the first bifurcation analysis for network models, channel or receptor conductance was gradually changed from $10^{-2}$ to $10^2$ times its original value by 17 steps (for the bifurcation by the pre-synaptic mechanism, $a_{AMPA}$

, $a_{xNMDA}$, $a_{GABA}$ were gradually changed from $10^{-2}$ to $10^1$ times its original value by 16 steps) and the simulations were conducted for 2.5 s. The membrane potential of all neurons in a network model were analyzed and classified into three categories based on the above criteria: "Sleep" (classified into "SWO" or "SWO with high frequency"), "Wake" (classified into "AWAKE" or "AWAKE with high frequency") and "Others" (classified into other categories).

For the evaluation of the degree of synchronization and desynchronization, coefficient of variance (CV) for total spike counts per 50 ms (sleep score) was calculated. More than 1,000 parameter sets that bifurcated from wake-like patterns as a network (defined by sleep scores <1.0 and the percentage of "Wake" neurons >30%) to wake-like patterns as a network (defined by sleep score ≥1.3 and the percentage of "Sleep" neurons >30%) were selected in each bifurcation model. Initial values in a network model were randomly assigned for $V$, $h$, $h_{Na}$, $n_K$, $h_A$ and $m_{KS}$ and other variables were the same as those of bifurcation analysis in a single neuron.

We conducted a second bifurcation analysis for the parameter sets collected by the first bifurcation analysis. In the second bifurcation analysis, the simulations were conducted for 5 s and the conductance was gradually changed from $10^{-2}$ to $10^{1.5}$ times its original value by 36 steps (for the bifurcation by the pre-synaptic mechanism, $a_{AMPA}$, $a_{xNMDA}$, $a_{GABA}$ were gradually changed from $10^{-2}$ to $10^{0.8}$ times its original value by 29 steps). The results were analyzed in the same way as the first bifurcation analysis and manually checked. The parameter sets including a stable sleep-like and wake-like firing patterns as a network with those mean firing rates that were close to each other (the difference is less than 2.0 Hz, called a representative sleep-like and wake-like state respectively) were selected.

We also constructed network models with different connection patterns because real connections of cortical networks are non-random lognormal-like connections [78]. The network models had different SDs and means of the lognormal distributions for the number of synapses per neuron from the original model (S10A and S10D Fig). Parameter sets that showed transitions from wake-like to sleep-like firing patterns after the first bifurcation analysis were selected. In the second bifurcation analysis, the ranges of the conductance or coefficient were from $10^{(-0.7)}$ to $10^{(sv^{+0.7})}$ times its original value by $(sv - wv + 1.4) \times 10$ steps ($wv$ and $sv$ were the values for the representative sleep-like and wake-like states as a network in the original model, respectively).

### Parameter search for synaptic learning rules in Hodgkin–Huxley-based network models

After the second bifurcation analysis, we analytically calculated the synaptic changes and searched for parameter sets representing specific learning rules in the same way as simple models. When using learning rules in network models, the equation (Eq. 4) was replaced by the following equation.

$$I_{Ca\_NMDA} = g_{NMDA} s_{NMDA} (V_{post} - V_{Ca})$$

In details, we selected a 4-s time series data for membrane potentials of a single neuron in a network model during wake-like states and duplicated the data with a short delay in each time lag (S6 Fig). Parameter sets for synaptic learning rules were randomly generated and the periods that post-synaptic $Ca^{2+}$ exceeded $\theta_p$ or $\theta_d$ in 3 s (from one to four seconds) were calculated respectively. Then, the periods spent above each threshold were multiplied by 20 to obtain the total values for 60 s for the purpose of speeding up the calculations, considering that the results would not change so much if time series data for membrane potentials were stable. Therefore, we selected time series data for membrane potentials of a representative neuron that had the minimum CV for firing rates in 1-s non-overlapping windows. Synaptic changes in each time lag were analytically calculated and the parameter sets of which the SSE between analytical results and the Gaussian fitting curves was less than 0.6 were selected. The ranges of $\theta_p$ and $\theta_d$ were limited to [0.8, 1.6] and [0.5, 1.0] when searching STDP and limited to [0.5, 1.0] and [0.8, 1.6] when searching Anti-STDP, respectively. The ranges of other parameters are shown in S2 Table. The same parameter sets for synaptic learning rules were applied to all excitatory synapses in a network model.

## Comparison of synaptic efficacy between sleep-like and wake-like firing patterns in Hodgkin–Huxley-based network models

A representative synaptic learning rule was obtained in each network model (see "Materials and methods, Parameter search for synaptic learning rules in Hodgkin–Huxley-based network models"). Time changes in synaptic efficacy were calculated for 60 s in each of wake-like and sleep-like patterns under specific learning rules in Fig 3. Scaling parameters ($\beta_{NMDA}$ and $\beta_{VGCC}$) and normalized thresholds for learning rules ($\theta_p$ and $\theta_d$) were calculated by preliminary simulations (see "Materials and methods, Obtaining values for scaling parameters and normalization of thresholds for synaptic learning rules"). The mean and CV of synaptic efficacy of excitatory neurons were calculated by every time step except for the initial 10 s and compared between sleep-like and wake-like patterns as an average over time. In this process, the network models with large differences of mean firing rates between sleep-like and wake-like patterns (>2.0 Hz) were excluded from the statistical analysis. Because the AMPAR activity in synapses reflects synaptic strength [79], the conductance of AMPAR was updated by synaptic efficacy ($\rho$).

$$g_{AMPA} = g_{AMPA\_min} + \rho \left( g_{AMPA\_max} - g_{AMPA\_min} \right) \tag{11}$$

$$g_{AMPA\_min} = g_{AMPA\_original} \times 0.5 \tag{12}$$

$$g_{AMPA\_max} = g_{AMPA\_original} \times 1.5 \tag{13}$$

## Calculation of synaptic efficacy under synaptic learning rules in Hodgkin–Huxley-based network models with sleep–wake dynamics

The equations of the two variable model for sleep–wake dynamics are based on the previous study [76].

$$\tau_r \frac{dr}{dt} = -r(t) + R_\infty \left( wr(t) + \alpha \left[ Ca^{2+} \right] - ba(t) + I + \xi \right) \tag{14}$$

$$R_\infty(x) = 1 / \left[ 1 + exp(-cx) \right] \tag{15}$$

$$\tau_a \frac{da}{dt} = -a(t) + A_\infty \left( r(t) - \beta \left[ Ca^{2+} \right] - d \right) \tag{16}$$

$$A_\infty(x) = 1 / \left[ 1 + exp(-ex) \right] \tag{17}$$

$$d\xi = -\theta\xi dt + \varepsilon\sqrt{2\theta dt}W_t \tag{18}$$

$$g_{NMDA} = g_{NMDA\_original} \times Max\_rate \times \left\{ a(t) \, or \, r(t) \right\} \tag{19}$$

*r* and *a* are the ratio of the first or second phosphorylated CaMKII. *w* is a coefficient of autoactivation and $\alpha$ is a coefficient of $Ca^{2+}$ activation for the first phosphorylated state of CaMKII. *b* is a coefficient of inhibition by the second phosphorylated CaMKII and $\beta$ is a coefficient of $Ca^{2+}$ inhibition for *a*. *I* is a constant input and $\xi$ is an input noise calculated by a Weiner process $W_t$, time scale $\theta$, and standard deviation $\varepsilon$. *r* and *a* approach their steady-state values defined by $R_\infty(x)$ *and* $A_\infty(x)$, respectively. *c*, *d*, and *e* are coefficients for $R_\infty(x)$ and $A_\infty(x)$. The conductance of NMDAR is updated by the value of *r* and *a*. We constructed the network of 80 neurons as in Fig 3 including synaptic learning rules and sleep–wake dynamics and calculated

the time change of *r* and **a**. Network models analyzed in Fig 3B–3F and with continuous transitions from wake-like patterns to sleep-like patterns were selected. For the model of sleep–wake dynamics bifurcated by the pre-synaptic mechanism, we assumed that the pre-synaptic Ca²⁺ ($[Ca^{2+}{}_{pre}]$) is mediated by VGCC and it was expressed by the following equations.

$$[Ca^{2+}{}_{pre}] = -\alpha_{Ca} A I_{Ca\_pre} \beta_{VGCC})$$

(20)

$$I_{Ca\_pre} = g_{ca} m^2{}_{Ca\infty} (V_{pre}) (V_{pre} - V_{Ca})$$

(21)

$[Ca^{2+}]$ in the equations (Eq. 14 and 16) was updated by Ca²⁺ in post-synaptic, intracellular or pre-synaptic compartments in the bifurcation model of the post-synaptic, intracellular or pre-synaptic mechanism, respectively. Sleep scores were calculated for 5-s windows and moving averages of them were computed on 10 consecutive values, which were classified into sleep-like and wake-like periods if sleep score was more than or less than the threshold, respectively (threshold is the value of sleep score when $p < 0.01$, see "Materials and methods, Evaluation of synchronization and desynchronization in Hodgkin–Huxley-based network models").

Then, we calculated Process S as following equations proposed by a previous study [80].

$$S_{t+1} = UA - (UA - S_t) \exp\left(-\frac{dt}{\tau_i}\right) \qquad (\textit{wake state})$$

(22)

$$S_{t+1} = LA + (S_t - LA) \exp\left(-\frac{dt}{\tau_d}\right) \qquad (\textit{sleep state})$$

(23)

where $S_t$ is the values of Process S, *UA* and *LA* are upper and lower asymptote, respectively. $\tau_i$ and $\tau_d$ are the time constant of the increasing and decreasing exponential saturating function, respectively.

We selected the models in which periods of sleep-like and wake-like states were close to each other (0.7 ≤ total sleep time/total wake time < 1.3) and Pearson's correlation coefficient between *r* or **a** and Process S was the largest. Scaling parameters ($\beta_{NMDA}$ and $\beta_{VGCC}$) and normalized thresholds for learning rules ($\theta_p$ and $\theta_d$) were calculated by preliminary simulations (see "Materials and methods, Obtaining values for scaling parameters and normalization of thresholds for synaptic learning rules"). The simulations were conducted for 300 or 500 s. The mean and CV of synaptic efficacy of all neurons were calculated by every time step during the periods of sleep-like and wake-like states and compared as an average over time.

## Obtaining values for scaling parameters and normalization of thresholds for synaptic learning rules

Values for scaling parameters ($\beta_{NMDA}$ and $\beta_{VGCC}$) were calculated when we compare synaptic efficacy between sleep-like and wake-like firing patterns in Hodgkin–Huxley-based network models according to the following procedures. $\beta_{NMDA}$ was set to be 1.0 and $\beta_{VGCC}$ was obtained by satisfying $2MC_{pre} = MC_{post}$ based on the results of preliminary simulations as conducted in simple model. $MC_{pre}$ and $MC_{post}$ were the mean amplitudes of $C_{pre}$ and $C_{post}$ in a 10-s simulation of wake-like patterns respectively, which were implemented in $\beta_{NMDA} = \beta_{VGCC} = 1.0$ and without learning rules (In the model with sleep–wake dynamics, $MC_{pre}$ and $MC_{post}$ were the mean amplitudes of $C_{pre}$ and $C_{post}$ during wake-like periods in simulations for 150 s in S11, S13 and S14 Figs or 250 s in Figs 4 and S12). $\theta_p$ and $\theta_d$ of synaptic learning rules were normalized by $MC_{pre}$ during wake-like patterns as the following equation.

$$\theta'_x = \frac{\theta_x \times MC_{pre}}{0.7} (x = p \ or \ d)$$

## Evaluation of synchronization and desynchronization in Hodgkin–Huxley-based network models

For the evaluation of the degree of synchronization and desynchronization, CV for total spike counts per 50-ms window without overlaps was calculated (Sleep score). Sleep scores were obtained from the waveforms of membrane potentials in all neurons for 2.5 s or 5.0 s in the first and second bifurcation analysis, respectively (Fig 3B–3F). In simulations with sleep–wake dynamics (Fig 4C–4F), sleep scores were calculated continuously in 5-s window with 500-ms overlaps. For the statistical tests, 1,000,000 sets of desynchronized spike trains for 2.5 s or 5 s (80 neurons, step size is 0.01) with the frequency of 0.5–15 Hz were randomly generated as the same way as in Fig 1G and 1H and sleep scores were calculated in each dataset. A sleep score of the data was compared with this distribution and regarded as sleep-like states if it was more than the value with $p < 0.01$. The distribution of sleep scores in generated desynchronized spike trains and sleep scores with $p < 0.01$ in different conditions are shown in S20 Fig.

## Calculation of synaptic efficacy under synaptic learning rules in Hodgkin–Huxley-based network models including stimulation during the wakefulness

The construction of the network model was the same as Figs 3 and 4. In S15 Fig, excitatory neurons were divided into three groups by their indices: group **1** and **2** (20 neurons in each group) are the groups for stimulated neurons and group **3** (24 neurons) is for unstimulated neurons. First, all neurons fire with desynchronized wake-like firing patterns. Then, neurons in group **1** and **2** were stimulated simultaneously at the end of wake-like states. The stimulation was optimized by changing its waveforms and rates so that the potentiation of synaptic efficacy between stimulated groups are observed under STDP during the stimulation protocol. After the stimulation, all neurons fire with synchronized sleep-like firing patterns. Mean of synaptic efficacy every time step and ratio of mean synaptic efficacy after to before sleep-like firing patterns were calculated and compared between stimulated neurons and unstimulated neurons.

## Statistical analysis

We applied a Student $t$ test in Figs 3E, 3F, S8, S9, S10B, S10C, S10E, S10F, S11, S16B and S16C. In Figs 1H, 2A–2C, 5, S2, S3, S16A, and S19, Bayesian statistics was applied to test the difference in mean synaptic efficacy and synaptic CV between two groups that follow normal distributions because the frequentist statistical tests with a large number of simulated results are likely to be significant. The analysis was performed according to the following steps: Non-informative priors were set for each group's mean and standard deviation and the likelihood for the observed data was defined. A parameter representing the difference in averages (diff = $mu_1 - mu_2$) ($mu_1$ and $mu_2$ is a mean in each group) was defined. Then, Markov Chain Monte Carlo method was used to sample from the posterior distribution. All posterior distributions were derived from 5,000 MCMC samples after 2,000 burn-in iterations. The sampling results were examined using 95% credible interval (95% CI) to evaluate whether there is a significant difference between the two groups (i.e., whether zero was contained within the interval).

## Software and numerical calculations

We used Python (version 3.8) with these following libraries: jupyter, matplotlib, numpy, pandas, scikit-learn, statannot, numba, seaborn, scipy, vistats, pymc, and arviz. All simulations were conducted using GPGPU (NVIDIA RTX 3,090, A5000 or A6000), C++17 and CUDA (version 12.0) except the simulations of parameter search for synaptic learning rules or parameter search for SWO and bifurcation analysis in a single neuron. Calculations were implemented by using a fourth-order Runge-Kutta method. Time steps were 0.1 ms in parameter search for synaptic learning rules, 0.05 ms in calculations of synaptic efficacy in simple model, and 0.00005–0.05 ms in calculations of synaptic efficacy in Hodgkin–Huxley-based network models, respectively.

## Supporting information

**S1 Fig. Distributions of parameter sets for synaptic learning rules in different fitting curves, related to** Fig 1. Distributions for parameter sets for four types of synaptic learning rules obtained by different fitting curves. A total of 1,000 parameter sets whose sum of squared errors (SSE) between the analytical result and fitted Gaussian curves was less than thresholds were selected. The threshold was 0.45 in $a_{lr} = 0.9$, $\tau_{lr} = 50$ **(E)** or 0.25 in other conditions **(A–D)**. **(A)** The original one. The plots in axes of thresholds and amplitudes were the same as in Fig 1F. **(B–E)** Parameter sets were collected by fitting to gaussian curves with different amplitudes ($a_{lr}$) and time constants ($\tau_{lr}$) from original one. The ranges of parameters are shown in S2 Table.
(TIF)

**S2 Fig. Mean synaptic efficacy at higher mean firing rates, related to** Fig 1. Box plots for mean synaptic efficacy under four different types of synaptic learning rules at higher mean firing rates ($n = 1,000$ for each firing rate, $n$ represents the number of synaptic learning rules). The parameter sets for synaptic learning rules were the same as in Fig 1H. The sleep-like firing patterns were generated by sampling from the lognormal distributions for Up-state duration and Down-state duration ($\log_{10}$ (*mean Up-state duration*) = 2.7 and $\log_{10}$ (*mean Down-state duration*) = 2.7, SD was calculated according to the linear regression analysis based on in vivo data (S18 Fig)). Initial synaptic efficacies in all the synapses were 0.5 and synaptic efficacies were simulated for 6 min. Synaptic efficacies for the last 2 min were averaged and compared between sleep-like and wake-like firing patterns. The whiskers above and below of box plots show minimal to maximal values. The box extends from the 25th to the 75th percentile and the middle line indicates the median. Bayesian statistical analysis was performed using Markov Chain Monte Carlo method to infer posterior distributions of average differences in mean synaptic efficacy between sleep-like firing patterns and wake-like firing patterns. Asterisks (*) indicate 95% CIs do not include zero. The data underlying the graphs shown in the figure can be found in Table A in S2 Data. The 95% CIs for the distributions of average differences are shown in Table F in S3 Data.
(TIF)

**S3 Fig. Mean synaptic efficacy in different fitting curves, related to** Fig 2. **(A–C)** Box plots for mean synaptic efficacy under synaptic learning rules in different parameters for fitting curves ($n = 1,000$ for each firing rate, $n$ represents the number of synaptic learning rules). The results of $a_{lr} = 0.7$, $\tau_{lr} = 30$ **(A)**, $a_{lr} = 0.5$, $\tau_{lr} = 50$ **(B)**, and $a_{lr} = 0.9$, $\tau_{lr} = 50$ **(C)** are shown. The parameter sets for synaptic learning rules were the same as in S1 Fig. The sleep-like firing patterns were the same as in Fig 1H. Initial synaptic efficacies in all the synapses were 0.5 and synaptic efficacies were simulated for 6 min. Synaptic efficacies for the last 2 min were averaged and compared between sleep-like and wake-like firing patterns. The whiskers above and below of box plots show minimal to maximal values. The box extends from the 25th to the 75th percentile and the middle line indicates the median. Bayesian statistical analysis was performed using Markov Chain Monte Carlo method to infer posterior distributions of average differences in mean synaptic efficacy between sleep-like firing patterns and wake-like firing patterns. Asterisks (*) indicate 95% CIs do not include zero. The data underlying the graphs shown in the figure can be found in Tables **B–D** in S2 Data. The 95% CIs for the distributions of average differences are shown in Tables G-I S3 Data.
(TIF)

**S4 Fig. Results of parameter search for SWO and bifurcation analysis, related to** Fig 3. **(A–F)** The results of parameter search for SWO and bifurcation analysis in single and multiple neurons of the Hodgkin–Huxley-based model is shown by three types of bifurcation models. The longitudinal axes represent the number of total parameter sets searched **(A)**, the percentage of parameter sets generating SWO **(B)**, the percentage of parameter sets bifurcating from wake-like to sleep-like firing patterns in a single neuron (bifurcation (single neuron)) **(C)**, the percentage of parameter sets bifurcating from wake-like to sleep-like firing patterns as a network (bifurcation (network)) **(D)**, ratio of bifurcation (single neuron) to SWO

**(E)**, ratio of bifurcation (network) to bifurcation (single neuron) **(F)**. The data underlying the graphs shown in the figure can be found in Table E in S2 Data.
(TIF)

**S5 Fig. Clustering of the results of bifurcation analysis, related to** Fig 3. Principal component analysis (PCA) was conducted for the parameter sets that bifurcated form sleep-like to wake-like firing patterns as a network obtained in bifurcation analysis in each Hodgkin–Huxley-based network model. **(A)** Cumulative contribution ratio of eigen values when PCA applied to the parameter sets. **(B)** Projection of the parameter sets onto their first two principal components in each model ($n$ = 1,344, 1,202, and 1,092 in the post-synaptic, intracellular and pre-synaptic bifurcation models, respectively).
(TIF)

**S6 Fig. Procedures of searching parameter sets for synaptic learning rules in Hodgkin–Huxley-based network model, related to** Fig 3. The details are shown in "Materials and methods, Parameter search for synaptic learning rules in Hodgkin–Huxley-based network models". A 4-s time series data for membrane potentials of a single neuron in a network model during wake-like states were selected and duplicated with a short delay in each time lag. Parameter sets for synaptic learning rules were randomly generated and the periods that post-synaptic $Ca^{2+}$ exceeded $\theta_p$ or $\theta_d$ in 3 s (from one to four seconds) were calculated respectively. Then, the periods spent above each threshold were multiplied by 20 to obtain the total values for 60 s for the purpose of speeding up the calculations. Synaptic changes in each time lag were analytically calculated and the parameter sets of which the SSE between analytical results and the Gaussian fitting curves was less than 0.6 were selected.
(TIF)

**S7 Fig. Time changes in membrane potential and synaptic efficacy of a representative network model under STDP, related to** Fig 3. **(A–E)** Time changes in membrane potential and synaptic efficacy were calculated in a representative intracellular bifurcation model. The structure and parameter set for the channel and receptor conductance of network model was the same as in Fig 3C and the parameter set for STDP learning rule is shown in S8 Table. Initial synaptic efficacies of all synapses were 0.5. Simulations were conducted for 60 s. Time changes in membrane potential during wake-like patterns **(A)**, membrane potential in sleep-like patterns **(B)**, synaptic efficacy of a representative synapse **(C)**, mean synaptic efficacy **(D)**, and CV of synaptic efficacy **(E)** are shown. Right figures show enlarged graphs for red rectangles of left figures.
(TIF)

**S8 Fig. Mean synaptic efficacy in Hodgkin–Huxley-based network models under Hebbian and Anti-Hebbian, related to** Fig 3. **(A, B)** Box plots for mean synaptic efficacy in sleep-like and wake-like firing patterns under Hebbian **(A)** and Anti-Hebbian **(B)** by three types of network models ($n$ = 192, 42 and 151 for Hebbian and $n$ = 143, 38 and 123 for Anti-Hebbian in post-synaptic, intracellular, and pre-synaptic bifurcation models respectively, $n$ represents the number of parameter sets for the network models). The structure and parameter set for channel or receptor conductance of network models were the same as in Fig 3E and 3F. A parameter set for the synaptic learning rule was assigned to each network model. Synaptic efficacy was compared assuming the almost close firing rates between sleep-like and wake-like states. Initial synaptic efficacies of all synapses were 0.5. Simulations were conducted for 60 s and synaptic efficacy and CV were averaged over the period from 10 to 60 s. The whiskers above and below of box plots show minimal to maximal values. The box extends from the 25th to the 75th percentile and the middle line indicates the median. *$p < 0.05$, **$p < 0.01$, ***$p < 0.001$, ****$p < 0.0001$, Student $t$ test was applied. The data underlying the graphs shown in the figure can be found in Table F in S2 Data.
(TIF)

**S9 Fig. Mean synaptic efficacy between sleep-like and wake-like firing patterns at different mean firing rates in Hodgkin–Huxley-based network models, related to** Fig 3. **(A, B)** Mean **(A)** and CV **(B)** of synaptic efficacy in the

post-synaptic bifurcation models under STDP were compared between sleep-like and wake-like firing patterns by different mean firing rates. The results of simulations for models bifurcated by the post-synaptic mechanism in Fig 3E were classified into three groups (0–4 Hz, 4–8 Hz and 8–12 Hz) by mean firing rates during wake-like firing patterns ($n = 35$, 106 and 29 for each firing rate group, $n$ represents the number of parameter sets for the network models). The box extends from the 25th to the 75th percentile and the middle line indicates the median. $*p < 0.05$, $**p < 0.01$, $***p < 0.001$, $****p < 0.0001$, Student $t$ test was applied. The data underlying the graphs shown in the figure can be found in Tables G and H in S2 Data.
(TIF)

**S10 Fig. Synaptic changes under STDP and Anti-STDP in Hodgkin–Huxley-based network models with different connections, related to Fig 3. (A)** Schematic illustration and histogram for lognormal distributions of the number of synapses per neuron in different SDs. **(B, C)** Box plots for mean synaptic efficacy during sleep-like and wake-like firing patterns under STDP ($n = 191, 74, 46, 43$, $n = 52, 34, 26, 25$, and $n = 150, 83, 56, 48$ for each value of SD in post-synaptic, intracellular and pre-synaptic bifurcation models, respectively) **(B)** and Anti-STDP ($n = 121, 82, 57, 49$, $n = 36, 34, 34, 31$, and $n = 119, 90, 61, 58$ for each value of SD in post-synaptic, intracellular and pre-synaptic bifurcation models, respectively) **(C)** in different SDs for lognormal distributions. **(D)** Schematic illustration and histogram for lognormal distributions of the number of synapses per neuron in different means. **(E, F)** Box plots for mean synaptic efficacy during sleep-like and wake-like firing patterns under STDP ($n = 89, 191, 85, 83$, $n = 36, 52, 34, 34$, and $n = 99, 150, 90, 88$ for each value of mean in post-synaptic, intracellular and pre-synaptic bifurcation models, respectively) **(E)** and Anti-STDP ($n = 83, 121,$ $85, 77$, $n = 36, 36, 35, 38$, and $n = 101, 119, 93, 91$ for each value of mean in post-synaptic, intracellular and pre-synaptic bifurcation models, respectively.) **(F)** in different means for lognormal distributions. **(B, C, E, F)** A parameter set for the synaptic learning rule was assigned to each network model. Synaptic efficacy was compared assuming the almost close firing rates between sleep-like and wake-like states. Initial synaptic efficacies of all synapses were 0.5. Simulations were conducted for 60 s and synaptic efficacy and CV were averaged over the period from 10 to 60 s. The whiskers above and below of box plots show minimal to maximal values. The box extends from the 25th to the 75th percentile and the middle line indicates the median. $*p < 0.05$, $**p < 0.01$, $***p < 0.001$, $****p < 0.0001$, Student $t$ test was applied. The parameter sets for channel and receptor conductance of network models and synaptic learning rules were the same as in Fig 3E and 3F. $n$ represents the number of parameter sets for the network models. The data underlying the graphs shown in the figure can be found in Tables I and J in S2 Data.
(TIF)

**S11 Fig. Analysis in multiple parameter sets for Hodgkin–Huxley-based network models bifurcated by the post-synaptic mechanism under STDP or Anti-STDP with sleep–wake dynamics, related to Fig 4. (A)** Pearson's correlation coefficients were calculated in combinations of $r$ and $a$ with multiple parameter sets for Hodgkin–Huxley-based network models. NMDAR conductance was updated by $r$ or $a$ and parameters for sleep–wake dynamics were optimized by Pearson's correlation coefficients between Process S and $r$ or $a$ (For example, $r$-$a$ means that conductance was updated by $r$ and parameters for sleep–wake dynamics were optimized by Pearson's correlation coefficients between Process S and $a$). The network structures and initial values for variables ($a$, $r$ and $\xi$) were the same as in Fig 4. Initial synaptic efficacies of all synapses were 0.5 and the simulations were conducted for 300 s. $n = 14, 14, 15,$ and 15 for $r$–$r$, $r$–$a$, $a$–$r$, and $a$–$a$ pair, respectively. $*p < 0.05$, $**p < 0.01$, $***p < 0.001$, $****p < 0.0001$, Student $t$ test was applied. The data underlying the graphs shown in the figure can be found in Table K in S2 Data. **(B, C)** Box plots for mean and CV of synaptic efficacy during sleep-like and wake-like periods in multiple parameter sets for network models with sleep–wake dynamics under STDP ($n = 27$) **(B)** and Anti-STDP ($n = 12$) **(C)**. Initial values for variables ($a$, $r$ and $\xi$) were the same as in the representative models in Fig 4. Initial synaptic efficacies of all synapses were 0.5 and the simulations were conducted for 300 s. NMDAR conductance was updated by $a$ and optimized by Pearson's correlation coefficients between Process S

and *r*. The whiskers above and below of box plots show minimal to maximal values. The box extends from the 25th to the 75th percentile and the middle line indicates the median. $*p < 0.05$, $**p < 0.01$, $***p < 0.001$, $****p < 0.0001$, Student *t* test was applied. The data underlying the graphs shown in the figure can be found in Tables L and M in S2 Data. (TIF)

**S12 Fig. Representative models with sleep–wake dynamics under STDP and Anti-STDP in the intracellular bifurcation model, related to** Fig 4. VGCC conductance was updated by *a* and parameters for the sleep–wake dynamics model were optimized by Pearson's correlation coefficient between Process S and *r*. Initial synaptic efficacies of all synapses were 0.5. The simulations were started from a sleep-like state and conducted for 500 s. The network structure was the same as in Fig 4B. The parameter set for channel or receptor conductance of network models, synaptic learning rules and sleep–wake dynamics and initial values for variables in a representative model are shown in S5–S8 Tables. **(A)** Schematic illustration of the model for sleep–wake dynamics in the intracellular bifurcation mechanism. $Ca^{2+}$ in a cell body activates the initial state of CaMKII represented by *r*. **(B, C)** Time changes in membrane potential of a neuron and post-synaptic $Ca^{2+}$, synaptic efficacy and ratio of two phosphorylated states of kinases (*r* and *a*) of a synapse in representative network models under STDP **(B)** and Anti-STDP **(C)**. The results from 200 to 500 s are shown. **(D, E)** Raster plots and time changes in mean synaptic efficacy, sleep score and Process S in representative network models under STDP **(D)** and Anti-STDP **(E)**. The shadow in time changes in mean synaptic efficacy represents SD. The network was considered to be in the sleep-like or wake-like states if the sleep score was above or below the threshold, respectively (the threshold is the value of sleep score where $p = 0.01$, see "Materials and methods, Evaluation of synchronization and desynchronization in Hodgkin–Huxley-based network models"). The results of 200–500 s are shown. **(F, G)** Mean and CV of synaptic efficacy during the periods of sleep-like and wake-like states in representative network models under STDP **(F)** and Anti-STDP **(G)**. The data underlying the graphs shown in the figure can be found in Table N in S2 Data. (TIF)

**S13 Fig. Representative models with sleep–wake dynamics under STDP and Anti-STDP learning rules in the pre-synaptic bifurcation model, related to** Fig 4. Coefficients for pre-synaptic activations were updated by *a* and parameters for the sleep–wake dynamics model were optimized by Pearson's correlation coefficient between Process S and *r*. Initial synaptic efficacies of all synapses were 0.5. The simulations were started from a wake-like state and conducted for 300 s. The network structure was the same as in Fig 4B. The parameter set for channel or receptor conductance of network models, synaptic learning rules and sleep–wake dynamics and initial values for variables in a representative model are shown in S5–S8 Tables. **(A)** Schematic illustration of the model for sleep–wake dynamics in the intracellular bifurcation mechanism. $Ca^{2+}$ in a pre-synaptic neuron activates the initial state of CaMKII represented by *r*. **(B, C)** Time changes in membrane potential of a neuron and post-synaptic $Ca^{2+}$, synaptic efficacy and ratio of two phosphorylated states of kinases (*r* and *a*) of a synapse in representative network models under STDP **(B)** and Anti-STDP **(C)**. **(D, E)** Raster plots and time changes in mean synaptic efficacy, sleep score and Process S in representative network models under STDP **(D)** and Anti-STDP **(E)**. The shadow in time changes in mean synaptic efficacy represents SD. The network was considered to be in the sleep-like or wake-like states if the sleep score was above or below the threshold, respectively (the threshold is the value of sleep score where $p = 0.01$, see "Materials and methods, Evaluation of synchronization and desynchronization in Hodgkin–Huxley-based network models"). **(F, G)** Mean and CV of synaptic efficacy during the periods of sleep-like and wake-like states in representative network models under STDP **(F)** and Anti-STDP **(G)**. The data underlying the graphs shown in the figure can be found in Table O in S2 Data. (TIF)

**S14 Fig. Sleep–wake dynamics in the bistable regime, related to** Fig 4. The model for sleep–wake dynamics can represent multiple regimes. In the oscillatory regime (Fig 4), activity alternates between sleep-like and wake-like states. In the bistable regime, although sleep-like and wake-like states are relatively stable, sufficiently large noises induces alternations

between two states, resulting in variable durations of two states. Synaptic efficacy was calculated in a representative network model bifurcated by the post-synaptic mechanism with sleep–wake dynamics and STDP learning rule in the bistable regime. The conductance of NMDAR was updated by *a* and the simulations were optimized by Pearson's correlation coefficients between Process S and *r*. Initial synaptic efficacies of all synapses were 0.5. The simulations were started from a wake-like state and conducted for 300 s. The network structure was the same as in Fig 4B. The parameter set for channel or receptor conductance of network models, synaptic learning rules and sleep–wake dynamics and initial values for variables in a representative model are shown in S5–S8 Tables. **(A)** Time changes in membrane potentials, post-synaptic Ca$^{2+}$, synaptic efficacy, and ratio of two phosphorylated states of kinases (*r* and *a*) in a single neuron of a representative network model. **(B)** Raster plots, time changes in mean synaptic efficacy, sleep score, and Process S in a representative network model. The shadow in time changes in mean synaptic efficacy represents SD. The network was considered to be in the sleep-like or wake-like states if the sleep score was above or below the threshold, respectively (the threshold is the value of sleep score where $p = 0.01$, see "Materials and methods, Evaluation of synchronization and desynchronization in Hodgkin–Huxley-based network models"). **(C)** Mean and CV of synaptic efficacy during sleep-like and wake-like periods in a representative network model. The data underlying the graphs shown in the figure can be found in Table P in S2 Data. **(D)** The results of simulations without noise. Simulations were conducted in $\theta = 0$ and $\xi = 0$ (the initial value for $\xi$ is also 0) while other parameters were the same as in simulations with noise. The upper graph shows the time change in sleep scores in the simulation starting from wake-like firing patterns and the lower graph shows the time change in sleep scores starting from the sleep-like firing patterns.
(TIF)

**S15 Fig. Synaptic changes in Hodgkin–Huxley-based network models with stimulation during wakefulness under STDP, related to** Fig 5**.** Synaptic efficacy was calculated in a representative network model bifurcated by the intracellular mechanism under STDP. Parameter sets for the channel or receptor conductance and the synaptic learning rule are shown in S5 and S8 Tables, respectively. The VGCC conductance was multiplied by $10^{-0.4}$ and $10^{-0.1}$ to its original value to generate wake-like and sleep-like firing patterns. Stimulations were applied for 15 s at 20 Hz after wake-like firing patterns. The stimulation was optimized by changing its waveforms and rates so that the potentiation of synaptic efficacy between stimulated groups were observed (see "Materials and methods, Calculation of synaptic efficacy under synaptic learning rules in Hodgkin–Huxley-based network models including stimulation during the wakefulness"). Initial synaptic efficacies of all synapses were 0.5. The network structure was the same as in Fig 4B. **(A)** Schematic illustration for grouping excitatory neurons. Group **1** and **2** were stimulated. **(B)** Time changes in mean synaptic efficacy of stimulated neurons and unstimulated neurons. **(C)** Ratio of mean synaptic efficacy after and before sleep-like firing patterns in stimulated and unstimulated neurons. The data underlying the graphs shown in the figure can be found in Table Q in S2 Data.
(TIF)

**S16 Fig. CV of synaptic efficacy under different synaptic learning rules. (A)** Box plots for CV of synaptic efficacy in sleep-like and wake-like firing patterns by synaptic learning rules and mean firing rates in simple model ($n = 1,000$ for each firing rate, *n* represents the number of synaptic learning rules). The network structure, firing patterns and parameters for synaptic learning rules are the same as in Fig 1H. Initial synaptic efficacies in all the synapses were 0.5 and synaptic efficacies were simulated for 6 min. Synaptic efficacies for the last 2 min were averaged and compared between sleep-like and wake-like firing patterns. The whiskers above and below of box plots show minimal to maximal values. The box extends from the 25th to the 75th percentile and the middle line indicates the median. Bayesian statistical analysis was performed using Markov Chain Monte Carlo method to infer posterior distributions of average differences in mean synaptic efficacy between sleep-like firing patterns and wake-like firing patterns. Asterisks (*) indicate 95% CIs do not include zero. The data underlying the graphs shown in the figure can be found in Table R in S2 Data. The 95% CIs for the distributions of average differences are shown in Table J in S3 Data. **(B, C)** Box plots for CV of synaptic efficacy during sleep-like

and wake-like firing patterns under STDP **(B)** and Anti-STDP **(C)** in Hodgkin–Huxley-based network models ($n$ = 191, 52 and 150 for STDP and $n$ = 121, 36 and 119 for Anti-STDP in post-synaptic, intracellular, and pre-synaptic bifurcation models respectively. $n$ represents the number of parameter sets for the network models). The network structure, firing patterns and parameters for synaptic learning rules are the same as in Fig 3E and 3F. Synaptic efficacy was compared assuming the almost close firing rates between sleep-like and wake-like states. Initial synaptic efficacies of all synapses were 0.5. Simulations were conducted for 60 s, and synaptic efficacy and CV were averaged over the period from 10 to 60 s. The whiskers above and below of box plots show minimal to maximal values. The box extends from the 25th to the 75th percentile and the middle line indicates the median. *$p < 0.05$, **$p < 0.01$, ***$p < 0.001$, ****$p < 0.0001$, Welch's $t$ test was applied in **(A)** and Student $t$ test was applied in **(B)** and **(C)**. The data underlying t*h*e graphs shown in the figure can be found in Table S in S2 Data.
(TIF)

**S17 Fig. Coefficient of noise and comparison of analytical and simulation results. (A)** The sum of squared errors (SSE) between analytical solutions and simulation results were calculated in different noise coefficient (a center figure) in simple model. All simulations were conducted in step size = 0.1. The simulation results were compared with analytical solutions and fitting curves (surrounding figures). The parameter set for STDP learning rule is shown in S8 Table. **(B)** Comparison of analytical and simulation results in parameter search for STDP learning rule in a representative Hodgkin–Huxley-based network model bifurcated by the post-synaptic mechanism. The parameter set for channel or receptor conductance of network model was the same as in Fig 3C and the parameter set for STDP learning rule is shown in S8 Table. The simulation was conducted by step size = 0.1.
(TIF)

**S18 Fig. Distributions of ISI and linear regression analysis for spike trains in vivo data. (A, B)** The distribution for ISI of the spike-train data for all excitatory neurons (verified by cross-correlogram) in a dataset of a previous article [8,22]. Lognormal distributions for ISI in Up states in the state of sleep **(A)** and wake **(B)** are shown. The lognormal distributions fitted well in the ISI of the state of wake **(B)**. Although mixed lognormal distributions were expected in the ISI of the sleep Up states, we assumed a single lognormal distribution in simulations for simplification **(A)**. **(C–G)** We performed the linear regression analysis on the mean and SD of Up-state duration **(C)**, the mean and SD of Down-state duration **(D)**, the mean and SD of ISI in the state of wake **(E)** and in the Up states of sleep **(F)** and the mean Up-state duration and mean Down-state duration **(G)**.
(TIF)

**S19 Fig. Mean synaptic efficacy in different membrane potential differences between Up and Down states. (A, B)** Box plots for mean synaptic efficacy during sleep-like and wake-like firing patterns with 10 mV **(A)** and 5 mV **(B)** membrane potential differences are shown ($n$ = 1,000 for each firing rate, $n$ represents the number of synaptic learning rules). The parameter sets for STDP and spike trains are the same as in Fig 1H. The parameters for constructing waveforms are shown in S3 Table. Initial synaptic efficacies in all the synapses were 0.5 and synaptic efficacies were simulated for 6 min. Synaptic efficacies for the last 2 min were averaged and compared between sleep-like and wake-like firing patterns. The whiskers above and below of box plots show minimal to maximal values. The box extends from the 25th to the 75th percentile and the middle line indicates the median. Bayesian statistical analysis was performed using Markov Chain Monte Carlo method to infer posterior distributions of average differences in mean synaptic efficacy between sleep-like firing patterns and wake-like firing patterns. Asterisks (*) indicate 95% CIs do not include zero. The data underlying the graphs shown in the figure can be found in Tables T and U in S2 Data. The 95% CIs for the distributions of average differences are shown in Table K in S3 Data.
(TIF)

**S20 Fig. Distribution of sleep scores and sleep scores with *p* < 0.01 in other conditions. (A)** Distribution of sleep scores in 1,000,000 desynchronized spikes in the condition of 80 neurons, 5 s simulation time and 0.5–15 Hz. Wake-like desynchronized spikes were sampled from lognormal distributions for ISI (see "Materials and methods, Definition of lognormal distributions based on in vivo recordings"). **(B–D)** sleep scores with *p* < 0.01 in different simulation times **(B)**, ranges of mean firing rates **(C)** and number of neurons **(D)**. *P* values were calculated by the distribution of 1,000,000 desynchronized spikes of each condition. The data underlying the graphs shown in the figure can be found in Table V in S2 Data.
(TIF)

**S1 Table. Fixed values in simple model for synaptic learning rules.** Values were based on the previous study [23] and used in Figs 1, 2, 5, S2, S3, S16A, S17A and S19.
(XLSX)

**S2 Table. Value ranges of parameters for synaptic learning rules.** Parameter values were randomly sampled from uniform distributions within these ranges of values when searching the parameter sets for synaptic learning rules fitting gaussian curves (Figs 1E, 1F, and S1).
(XLSX)

**S3 Table. Parameters for constructing voltage waveforms from spike trains.** These are values used in Figs 1, 2, 5, S2, S3, S16A, S17A and S19 to construct voltage waveforms from spike trains.
(XLSX)

**S4 Table. Fixed values in the Hodgkin–Huxley-based network model.** Values were based on the previous study [23] and used in simulations of Hodgkin–Huxley-based network models.
(XLSX)

**S5 Table. The representative parameter sets for Hodgkin–Huxley-based network models.** These parameter sets were used in the network models by three types of bifurcations in Figs 3C and 4, S7, and S12–S17.
(XLSX)

**S6 Table. The representative parameter sets for sleep–wake dynamics S7 Table.** Initial values in the representative models with sleep–wake dynamics. These parameter sets were used in simulations of the Hodgkin–Huxley-based network models with sleep–wake dynamics in Figs 4 and S12–S14.
(XLSX)

**S7 Table. Initial values in the representative models with sleep–wake dynamics.** The initial values for channel or receptor conductance, or a coefficient of pre-synaptic activation and variables of sleep–wake dynamics were as follows.
(XLSX)

**S8 Table. The representative parameter sets for synaptic learning rules.**
(XLSX)

**S1 Data. The data underlying the graphs shown in Figs 1–5.**
(XLSX)

**S2 Data. The data underlying the graphs shown in S2–S4, S8–S16, S19, and S20 Figs.**
(XLSX)

**S3 Data. The 95% credible intervals for distributions of average differences between sleep-like and wake-like firing patterns calculated by Bayesian statistical analysis in Figs 1H, 2A–2C, 5, S2, S3, S16A, and S19.** (XLSX)

## Acknowledgments

We thank all the laboratory members at RIKEN Center for Biosystems Dynamics Research and the University of Tokyo. We thank M. Graupner for guidance of codes. We thank B. O. Watson and D. Levenstein for guidance of in vivo datasets. We thank MN Ballester Roig for reviewing manuscripts. We thank G. Buzsáki, V. V. Vyazovskiy, C. Cirelli, G. H. Diering, P. Meerlo, P. Franken, M. G. Frank, A. Adamantidis, H. C. Heller, M. Schmidt and A. Loudon for helpful discussions.

## Author contributions

**Conceptualization:** Fukuaki L. Kinoshita, Rikuhiro G. Yamada, Hiroki R. Ueda.

**Funding acquisition:** Hiroki R. Ueda.

**Investigation:** Fukuaki L. Kinoshita.

**Methodology:** Fukuaki L. Kinoshita, Rikuhiro G. Yamada, Koji L. Ode, Hiroki R. Ueda.

**Project administration:** Hiroki R. Ueda.

**Resources:** Rikuhiro G. Yamada, Koji L. Ode.

**Software:** Fukuaki L. Kinoshita.

**Supervision:** Hiroki R. Ueda.

**Validation:** Fukuaki L. Kinoshita, Rikuhiro G. Yamada, Koji L. Ode.

**Visualization:** Fukuaki L. Kinoshita.

**Writing – original draft:** Fukuaki L. Kinoshita.

**Writing – review & editing:** Fukuaki L. Kinoshita, Rikuhiro G. Yamada, Koji L. Ode, Hiroki R. Ueda.

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
