## [Editor Report · Decision Letter 0]

30 Jan 2025

Dear Dr Ueda, 

Thank you for submitting your manuscript entitled "Boundary conditions for synaptic homeodynamics during the sleep-wake cycle" for consideration as a Research Article by PLOS Biology, and please accept my apologies for the delay in sending you an initial decision on your manuscript. We understand that your study has been revised in response to reviewer comments provided by Review Commons and we have now had a chance to discuss your paper, the reviews, and your response to reviewers with an academic editor with relevant expertise. 

We are in principle, interested in your study and would like to send your submission out for external peer review. However, I wanted to let you know that we think that it will be important to add a fourth reviewer, with strong computational modelling expertise, to complement the expertise of the reviewers from Review Commons and help assess the validity of the model. There is a chance that the fourth reviewer would identify new issues, and we would need to take those into account. 

If that is OK with you, before we can send your manuscript to reviewers, we need you to complete your submission by providing the metadata that is required for full assessment. To this end, please login to Editorial Manager where you will find the paper in the 'Submissions Needing Revisions' folder on your homepage. Please click 'Revise Submission' from the Action Links and complete all additional questions in the submission questionnaire.

Once your full submission is complete, your paper will undergo a series of checks in preparation for peer review. After your manuscript has passed the checks it will be sent out for review. To provide the metadata for your submission, please Login to Editorial Manager (https://www.editorialmanager.com/pbiology) within two working days, i.e. by Feb 03 2025 11:59PM.

Kind regards,

Luke

Lucas Smith, Ph.D.

Senior Editor

PLOS Biology

lsmith@plos.org

---

## [Decision Letter · Decision Letter 1]

24 Mar 2025

Dear Dr Ueda,

Thank you for your patience while we considered your revised manuscript "Boundary conditions for synaptic homeodynamics during the sleep-wake cycle" for consideration as a Research Article at PLOS Biology. Your revised study has now been evaluated by the PLOS Biology editors, the Academic Editor and the original reviewers.

In light of the reviews, which you will find at the end of this email, we are pleased to offer you the opportunity to address the remaining points from the reviewers in a revision that we anticipate should not take you very long. We will then assess your revised manuscript and your response to the reviewers' comments with our Academic Editor aiming to avoid further rounds of peer-review, although we might need to consult with the reviewers, depending on the nature of the revisions.

Please also make sure to address the following data and other policy-related requests:

* We would like to suggest a different title to improve its accessibility for our broad audience: "A unified framework to model synaptic dynamics during the sleep-wake cycle"

* Please add the links to the funding agencies in the Financial Disclosure statement in the manuscript details.

* DATA POLICY:

Regardless of the method selected, please ensure that you provide the individual numerical values that underlie the summary data displayed in the following figure panels as they are essential for readers to assess your analysis and to reproduce it: 1H, 2ABC, 3EF, 4GH, 5, S2, S3, S4, S8, S9, S10, S11, S12FG, S13FG, S14, S15, S16, S19 and S20. 

* CODE POLICY

* Please note that per journal policy, the model system/species studied should be clearly stated in the abstract of your manuscript. 

* Please move the supplementary methods and references to the main manuscript file. 

**IMPORTANT - SUBMITTING YOUR REVISION**

*Resubmission Checklist*

*Published Peer Review*

*PLOS Data Policy*

*Blot and Gel Data Policy*

Sincerely,

Christian

Christian Schnell, Ph.D.

Senior Editor

PLOS Biology

cschnell@plos.org 

on behalf of 

Lucas Smith, Ph.D.

Senior Editor

PLOS Biology

lsmith@plos.org

REVIEWS:

Reviewer #1: In this manuscript, Kinoshita and colleagues present a new modelling approach and framework in which to understand the relationships among sleep/wake states, neuronal activity, rules for learning, and synaptic plasticity. Through their modelling of Hebbian and spike-timing dependent plasticity, they identified a tendency, wake inhibition and sleep excitation (WISE) in which firing patterns associated with wake decreased synaptic weights, while patterns of neuronal firing that resembled sleep strengthened synaptic weights. This differed from anti-Hebbian and anti-spike timing dependent plasticity in which synaptic depression was modelled under NREM sleep. This latter observation is akin to the synaptic homeostasis hypothesis. They further showed that these synaptic changes depended on firing rate differences between NREM and wakefulness. Thus, there is considerable potential with this model/approach that could lead to a big change in how synaptic plasticity, neuronal activity, and learning are thought off. 

The manuscript is much improved and the authors have attempted to address many of points raised previously. The manuscript is improved by the authors altering their manuscript. There are still some questions that I think merit consideration. 

1) It is not entirely clear why the authors have assumed that CamKII is responsible for the homeostatic oscillation. This needs to be better fleshed out. Are there other kinases or signaling pathways that are of similar importance? How do the authors incorporate the different phosphorylation states of these enzymes? Similarly, it is not too obvious as to why the second state interacts through molecules that induce slow wave sleep oscillation.

2) I am somewhat confused by the role of firing rate in these processes. In their rebuttal to reviewer 1, they emphasized that the mechanism via which firing patterns are generated is important. It could be argued that this also alters the types of neurochemicals being released as burst firing is often associated with peptide or hormone release which does not seem to be integrated into the model or types of synaptic plasticity. This should be expanded upon in the discussion.

3) The range of duration in up and downstates across species is broad but it is difficult to see how the authors have validated their model to support these assumptions. Please expand.

Minor point:

Page 4, line 86. Replace 'Reverse types of them...' with 'Reverse types of these...' and replace These at the start of the next sentence (line 87) with Such. 

Reviewer #2: Kinoshita et al

I was not one of the original reviewers of the paper, so I feel I should not be too critical to avoid 'double jeopardy' for the authors.

Overall, the study seems competently carried out. I think without further determination of in vivo plasticity rules under awake and sleep states, the conclusions remain a bit tentative, but they set a decent baseline for subsequent studies.

- the authors should be more explicit about the brain area they model.

- Fig1, what is the mean efficacy before stimulation?

- I should also note that formally, the logic of drawing many models is slightly flawed.

Afterall, biology could have choosen a very particular setting, so that the outcome is not at all well described by the mean of many random choices.

- there are many cases where variables are not named. This should be corrected and every variable should be defined (for an example see Hodgkin Huxley 1952)

The section on NMDA is particularly confusing. x_nmda has a DE but it is not used above it which uses x. But what even is it? It has a remarkably short time-scale.

If it is really that short, it could be omitted and the value replaced with its steady state.

The section on homeostasis (l444) is also particularly poor.

- l132. Why is this called a simplified model? In the network in Fig2B, which neurons received the external input?

- if the authors continue this work, they might be interested in:

A stochastic model of hippocampal synaptic plasticity with geometrical readout of enzyme dynamics YE Rodrigues, CM Tigaret, H Marie, C O'Donnell, R Veltz Elife 12, e80152

- I'm not a proponent of using t-test for modelling results (after all, once one runs sufficient simulations, everything, apart from trivial cases, will be significant...)

---

## [Editor Report · Decision Letter 2]

5 May 2025

Dear Hiro,

Thank you for the submission of your revised Research Article "A unified framework to model synaptic dynamics during the sleep-wake cycle" for publication in PLOS Biology and apologies for my delay in sending you a decision. I was at a conference when your paper came in, and so it too me a bit longer than normal to look at your revision. Your responses to our editorial requests and the reviewer comments from the last round of review have now been assessed by the PLOS Biology editorial team and the Academic Editor, Jozsef Csicsvari. On behalf of my colleagues, I am pleased to say that we are satisfied by the changes made, and can in principle accept your manuscript for publication, provided you address any remaining formatting and reporting issues. These will be detailed in an email you should receive within 2-3 business days from our colleagues in the journal operations team; no action is required from you until then. Please note that we will not be able to formally accept your manuscript and schedule it for publication until you have completed any requested changes.

PRESS

Sincerely, 

Luke

Lucas Smith, Ph.D.

Senior Editor

PLOS Biology

lsmith@plos.org